# High-throughput discovery of functional disordered regions: investigation of transactivation domains

Charles NJ Ravarani[1,*] (iD), Tamara Y Erkina[2], Greet De Baets[1], Daniel C Dudman[2], Alexandre M Erkine[2,**] (iD) & M Madan Babu[1,***] (iD)

## Abstract

**Over 40% of proteins in any eukaryotic genome encode intrinsically disordered regions (IDRs) that do not adopt defined tertiary structures. Certain IDRs perform critical functions, but discovering them is non-trivial as the biological context determines their function. We present IDR-Screen, a framework to discover functional IDRs in a high-throughput manner by simultaneously assaying large numbers of DNA sequences that code for short disordered sequences. Functionality-conferring patterns in their protein sequence are inferred through statistical learning. Using yeast HSF1 transcription factor-based assay, we discovered IDRs that function as transactivation domains (TADs) by screening a random sequence library and a designed library consisting of variants of 13 diverse TADs. Using machine learning, we find that segments devoid of positively charged residues but with redundant short sequence patterns of negatively charged and aromatic residues are a generic feature for TAD functionality. We anticipate that investigating defined sequence libraries using IDR-Screen for specific functions can facilitate discovering novel and functional regions of the disordered proteome as well as understand the impact of natural and disease variants in disordered segments.**

**Keywords** high-throughput screen; intrinsically disordered protein; machine learning; mutational scanning; transactivation domain
**Subject Categories** Genome-Scale & Integrative Biology; Structural Biology; Transcription
**Mol Syst Biol. (2018) 14: e8190**

## Introduction

Understanding how the amino acid sequence of a protein contributes to its function (sequence–function relationship) is a problem of long-standing interest. The work of Anfinsen and colleagues in the 1960s (Anfinsen, 1973) together with the elucidation of protein structures established the sequence–structure–function paradigm (Fersht, 2008). With the availability of genomes, it has become clear that a large fraction of any eukaryotic proteome encodes protein segments that do not autonomously fold into a defined tertiary structure although they may contain secondary structural elements (van der Lee *et al*, 2014). Proteins typically use their intrinsically disordered regions (IDRs) to perform their function by mediating transient protein interactions (Tompa *et al*, 2014; Van Roey *et al*, 2014). Such regions can tolerate mutations; hence, they evolve rapidly and acquire functionality through both convergent evolution and divergent evolution (van der Lee *et al*, 2014; Tompa *et al*, 2014; Davey *et al*, 2015). Although computational approaches have estimated that there could be up to a million functional IDRs in the human proteome (Tompa *et al*, 2014), only a small fraction of them have been characterized so far (Gouw *et al*, 2017), limiting our understanding of the disordered proteome.

*In vitro* technologies such as phage display are powerful to identify short disordered linear motifs (three to seven residues within IDRs) that can mediate interactions with specific protein domains *in vitro* as well as discover strong binders (Ivarsson *et al*, 2014; Garrido-Urbani *et al*, 2016; Davey *et al*, 2017). Such approaches require screening short peptides against specific interaction partners, thus constraining the mechanism by which they mediate function (Jones *et al*, 2006; Ivarsson *et al*, 2014; Garrido-Urbani *et al*, 2016; Davey *et al*, 2017). The screening occurs outside of the relevant cellular/biological context during the selection experiment and hence does not explicitly consider cellular specificity for binding, i.e., selection against promiscuous binding with other molecules in the cell (negative selection). Thus, there is a need for a complementary and systematic high-throughput approach to study the sequence–function relationship of IDRs in a biologically relevant cellular context.

We present a framework called IDR-Screen that allows mechanism-independent discovery of disordered regions that are functional in a cellular context (Fig 1). It leverages various techniques, including mutational scanning of pooled sequences, genetic screens, and machine learning (ML; Boucher *et al*, 2014; Fowler & Fields, 2014; Jordan & Mitchell, 2015; Geffen *et al*, 2016; Nim *et al*, 2016; Rocklin *et al*, 2017), and consists of the following modular steps: (i) designing and generating libraries of sequences that code for short peptide segments, (ii) generating a cell population carrying the

1  MRC Laboratory of Molecular Biology, Cambridge, UK
2  Butler University, Indianapolis, IN, USA
   *Corresponding author. Tel: +44 1223 267836; E-mail: ravarani@mrc-lmb.cam.ac.uk
   **Corresponding author. Tel: +1 317 940 8569; E-mail: aerkine@butler.edu
   ***Corresponding author. Tel: +44 1223 267066; E-mail: madanm@mrc-lmb.cam.ac.uk

different sequences and screening them for a function of interest using a selection system (e.g., based on cell viability), (iii) sequencing the population before and after selection and determining functional and non-functional sequences, (iv) describing all sequences by calculating a feature vector quantifying their molecular properties, and (v) applying data analysis approaches such as ML to highlight the molecular basis of functionality of the short disordered peptides in the library (Fig 1). Here, we study transcription initiation as a model biological process to discover and learn what makes certain disordered regions functional (Appendix Figs S1–S3).

## Results

### High-throughput screening of random sequence library for transactivation domain discovery

In addition to the DNA binding domain that binds to the promoter DNA, transcription factors (TFs) also harbor transactivation domains (TAD), which are typically less than 20 residues and intrinsically disordered (Sigler, 1988). The current mechanistic model is that TAD mediates interactions to recruit the transcriptional machinery, which is critical for transcription initiation (Ptashne & Gann, 1997). Early investigations of individual TADs of TFs as well as screens of random DNA sequences and *Escherichia coli* genomic fragments have revealed that TADs tend to be disordered (i.e., unstructured; Sigler, 1988), enriched for acidic (Ma & Ptashne, 1987; Erkine & Gross, 2003) and hydrophobic residues (Cress & Triezenberg, 1991; Regier *et al*, 1993; Drysdale *et al*, 1995; Lu *et al*, 2000; Erkine & Gross, 2003), have a propensity to form alpha helices (i.e., intrinsic helicity) upon binding to their interaction partner (Uesugi *et al*, 1997; Lee *et al*, 2010) and may contain distinct sequence motifs that mediate interactions with specific components of the transcriptional machineries (Kussie *et al*, 1996; Radhakrishnan *et al*, 1997; Jonker *et al*, 2005; Piskacek *et al*, 2007). This has led to TADs being referred to as "acid blobs and negative noodles" (Sigler, 1988). While most TADs are enriched for these properties, the set of features above do not robustly define a TAD sequence when considered individually (Abedi *et al*, 2001; Bhaumik & Green, 2001; Mapp & Ansari,

## IDR-Screen - Discovery of functional disordered regions and learning what makes them functional

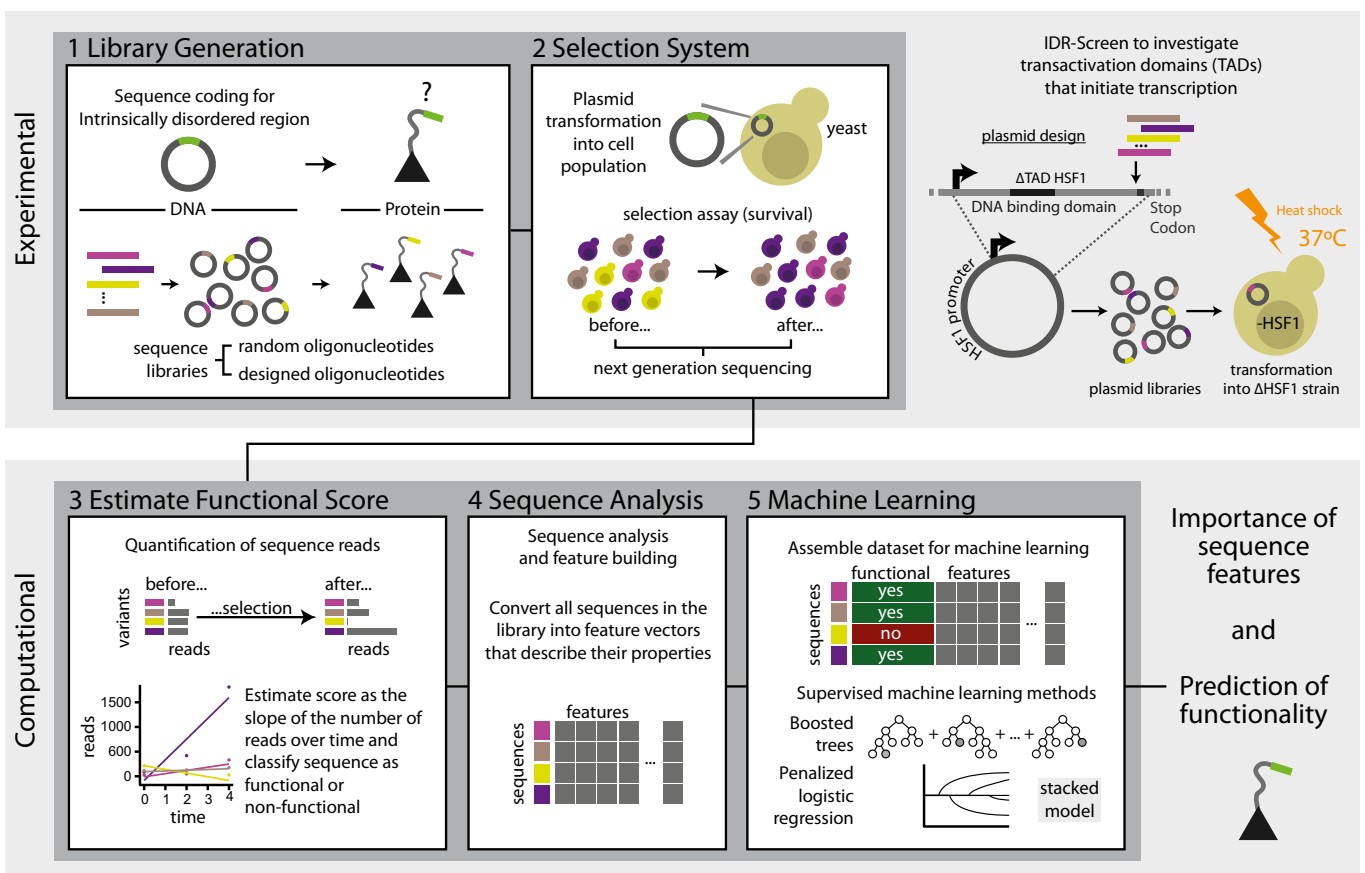

**Figure 1.  Outline of IDR-Screen.**
IDR-Screen consists of a modular set of stages that can broadly be grouped into the experimental and computational phases. A library of random or designed sequences is transformed into a cell population, expressed as a part of a protein that is used for selection (survival or other readouts such as fluorescence). In this manner, the library is screened to discover sequences that are functional/non-functional based on the designed assay. Upon data processing, this dataset of experimentally validated functional and non-functional sequences are analyzed to learn the rules of functionality using machine-learning (ML) approaches.

    

2007; Hahn & Young, 2011; Warfield *et al*, 2014; Erkina & Erkine, 2016).

To discover which sequences can function as TADs, we investigated a library of random DNA sequences (60 bp, ≤ 20aa; random library). Since the encoded peptide sequences are ≤ 20aa, such segments may contain secondary structures of varying degrees but are unlikely to form defined tertiary structures (Murzin *et al*, 1995) and hence more likely to be disordered. Different selection assays can be designed to discover TADs. We developed an assay to discover functional sequences using the yeast heat shock factor 1 (HSF1) transcription factor as our model. HSF1 has several functional regions including a DNA binding domain, a trimerization domain, and a disordered segment containing a C-terminal TAD (Morimoto, 1998) and regulates the expression of several genes to launch a heat shock response (Hahn *et al*, 2004; Appendix Fig S1A). Deletion of the disordered C-terminal TAD (HSF1-ΔTAD) results in cell death when grown at 37°C (Erkine & Gross, 2003; Sorger, 1990; Appendix Fig S1B). We then fused the library of sequences to HSF1-ΔTAD and subjected them to the selection experiment. We inferred that sequences that confer survival at 37°C have the potential to function as TADs in this biological context. On the other hand, sequences that mediate promiscuous interactions or fail to initiate transcription efficiently will eventually drop out of the screen. Thus, non-functional sequences will negatively affect growth of cells harboring them or result in cell death (Appendix Fig S1C and D). In this manner, the assay design incorporates the relevant cellular context and negative selection.

## Functional sequences display sequence property enrichments compared to non-functional ones

Using this assay, we obtained robust measurements for 67,263 random sequences (i.e., transformed and detected in at least two replicate experiments; Materials and Methods; Table EV1). Using stringent criteria to ensure a low false-positive rate (Materials and Methods), we identified 739 sequences (~ 1%) that confer survival and hence could function as TADs. Representative sequences from this screen were independently sequenced and confirmed to confer TAD functionality through spot-dilution assay experiments (Appendix Fig S4). An advantage of the IDR-Screen approach is that in addition to discovering functional sequences, the non-functional sequences that are experimentally validated through the screen (with negative selection considerations) provide a more appropriate control set of sequences to compare against. Functional sequences show enrichment for negatively charged residues (D, E) as well as aromatic amino acids (F, W, Y), compared to the non-functional sequences (Fig 2A). Furthermore, functional sequences were depleted in positively charged residues (R, K, and H). In terms of sequence properties, the functional sequences tend to be longer (median length: 18 residues), have a lower isoelectric point (median pI: 5.57), higher hydrophobicity (median % hydrophobicity: 0.33), intermediate propensity to be disordered (median probability: 0.57), and display some helical propensity (median % helicity: 0.14). The functional sequences are also enriched for the occurrence of certain linear peptide motifs (9-amino acid TAD; Piskacek *et al*, 2007; Fig 2B–G).

We then assessed the predictive power of these individual sequence properties in discriminating functional from non-functional sequences. Given the imbalance in our dataset (739 functional: 63,385 non-functional; imbalance ratio: 0.0117), we used sub-sampling when training the models and assessed the performance using precision–recall curves (Materials and Methods). Using logistic regression models, we find that the aforementioned properties such as length, pI, hydrophobicity, disorder content, intrinsic helicity, and the occurrence of a 9-aa TAD motif poorly discriminate functional and non-functional sequences in the random library when considered individually (Appendix Fig S5A and Materials and Methods). In other words, several sequences that do not function as TADs are frequently erroneously predicted to be functional when only these properties were considered and a number of functional sequences will be often incorrectly predicted to be non-functional (see Appendix Fig S5B–G for examples). Among all the properties tested, the pI of the sequence appears to have the most discriminative power. We then combined these sequence properties in our model (rather than consider them individually), which marginally affected the ability to discriminate the sequences (Appendix Fig S6A and B). Thus, the properties described above do not exhaustively describe TAD functionality, suggesting that a more exhaustive set of features could increase the predictive power to discriminate functional and non-functional sequences.

## Machine learning provides a robust approach to assess sequence feature importance

We therefore developed several different features that more comprehensively describe every sequence in the library in addition to the previously described ones (Appendix Fig S2; Table EV2). Analysis of the functional sequences showed prevalence of short, highly degenerate motifs (~ 2–5 residues in length involving negative and aromatic residues; Appendix Fig S7). It is known that tryptophan residues can stabilize local conformations of protein segments (Cochran *et al*, 2001) and is important to mediate interactions with other proteins as in the case of EIF3 (Marcotrigiano *et al*, 1997). We designed several new features that captured chemical properties as well as patterns of spacing such as the combinations of amino acids of defined properties (e.g., aromaticity, aliphaticity, hydrophobicity, and presence of positively and negatively charged residues) that are separated by defined distances in the sequences (degenerate mini-motifs). In this manner, we computed 146 different features that were broadly grouped into eight general feature sets (Materials and Methods; Table EV2). We then minimized feature redundancy by retaining one of the highly correlated features after clustering them based on similarity (Materials and Methods). To ensure robust analysis and effective interpretation of patterns in the data, we used different algorithms that rely on distinct principles and provide feature importance for interpretation of the models. Specifically, we used ML algorithms that assume linear (penalized logistic regression model; lasso and ridge) and nonlinear (boosted tree model) relationship between the features for classification (Materials and Methods). Both of these approaches have intrinsic feature ranking capacities. Using the features described above, we trained the different models, each scanning over a broad range of parameters (James *et al*, 2013). We also employed a stacked model that combines the best models from these approaches (Materials and Methods) and considers them together when identifying the feature importance.

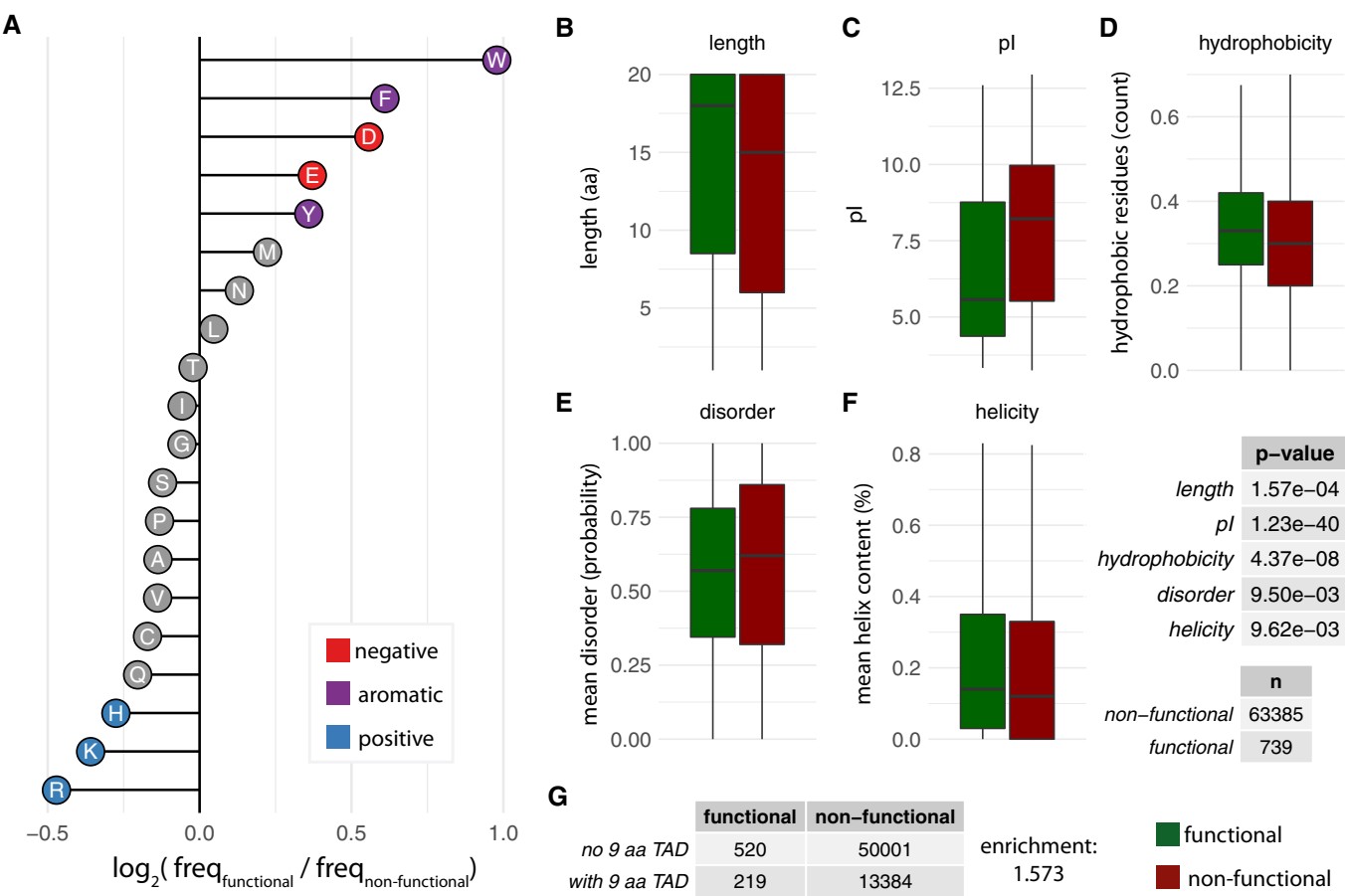

**Figure 2. Analysis of functional and non-functional sequences from the random library.**

A    Enrichment and depletion of different amino acids in the random library (log2 of frequencies of functional over non-functional sequences).

B–G    Boxplots of the distribution of the values of length (B), pI (C), hydrophobicity (D), disorder content (E), and helicity (F) for sequences that are functional (green) and non-functional (red). In the boxplots, the central line shows the median. Statistical significance was assessed using Wilcoxon test, *n* values (sample size) and *P*-values are provided on the right. (G) Enrichment of 9-aa TAD motif in functional versus non-function sequences; ratio of with-to-without 9-aa TAD in functional-to-non-functional sequences (219/520)/(13384/50001).

Although the predictive power of the ML models is not high, they identify features that make sequences functional (Appendix Fig S8A and B; for the best performing models: precision–recall (PR) area under the curve (AUC): 0.0563; random performance: 0.0115 and receiver operator curve (ROC) AUC: 0.6875; random performance: 0.5). It is also a way to test different hypothesis through the importance of a specific feature. Since all features are tested together while training the model, the ranking based on feature importance highlights their relative importance in discriminating functional from non-functional sequences. We highlight features that contributed the most to predict a functional sequence in our random library in the different ML models (Fig 3; Table EV3). One of the key features in the top 10 that contributed the most includes a degenerate mini-motif with a prevalence of negatively charged residues (enriched: D, E) in proximity to aromatic (enriched: F, W, Y) amino acids. Other feature sets that are important include the pI, single amino acid composition (enriched: W, D, N; depleted: S), and grouped amino acid composition (enriched: aromatic, hydrophobic, and negative; depleted: polar). We also tested the 9-aa TAD motif and helicity among others, which were not among the top 10

features. This highlights that functional sequences need not be restricted to contain specific sequence motifs or show a tendency to form specific secondary structure elements. These observations suggest that IDRs that contain multiple degenerate mini-motifs of negatively charged and aromatic residues, and devoid of positively charged residues are a generic descriptor of functional sequences in the random library. Consistent with this, an analysis of available structures of TADs in complex with their interaction partner in the Protein Data Bank (Rose *et al*, 2017) revealed a common theme for the known TAD interacting domains, which tend to contain a positively charged patch and a hydrophobic binding pocket (Appendix Fig S9).

## Studying naturally occurring TADs and their variants provides insights into functional features

The IDR-Screen approach can also be used to investigate libraries of naturally occurring sequences and designed variants to probe and learn about naturally occurring TADs. To highlight this, we investigated 25 transactivation domains of TFs and focused on 13

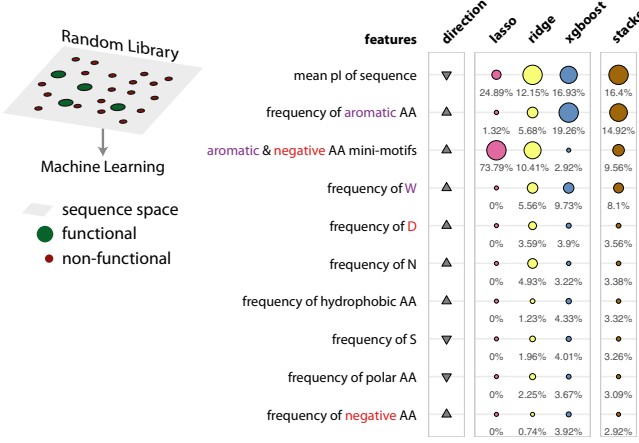

**Figure 3. The top 10 most important features of the machine-learning models trained on the random library.**

Schematic describing the sequence space explored by the random library (left). Table listing the top 10 most important features. The relative feature importance is given as relative percentages in the last four columns. The size of the circles is scaled per method (lasso, ridge, xgboost, stacked). The direction column denotes the direction of enrichment of the given feature for functional sequences compared to non-functional sequences (up, positive direction and down, negative direction). This figure provides a simplified description of the actual features, which are available in Table EV3.

| features | direction | lasso | ridge | xgboost | stacked |
|---|---|---|---|---|---|
| mean pI of sequence | ▽ | 24.89% | 12.15% | 16.93% | 16.4% |
| frequency of aromatic AA | ▲ | 1.32% | 5.68% | 19.26% | 14.92% |
| aromatic & negative AA mini-motifs | ▲ | 73.79% | 10.41% | 2.92% | 9.56% |
| frequency of W | ▲ | 0% | 5.56% | 9.73% | 8.1% |
| frequency of D | ▲ | 0% | 3.59% | 3.9% | 3.56% |
| frequency of N | ▲ | 0% | 4.93% | 3.22% | 3.38% |
| frequency of hydrophobic AA | ▲ | 0% | 1.23% | 4.33% | 3.32% |
| frequency of S | ▽ | 0% | 1.96% | 4.01% | 3.26% |
| frequency of polar AA | ▽ | 0% | 2.25% | 3.67% | 3.09% |
| frequency of negative AA | ▲ | 0% | 0.74% | 3.92% | 2.92% |

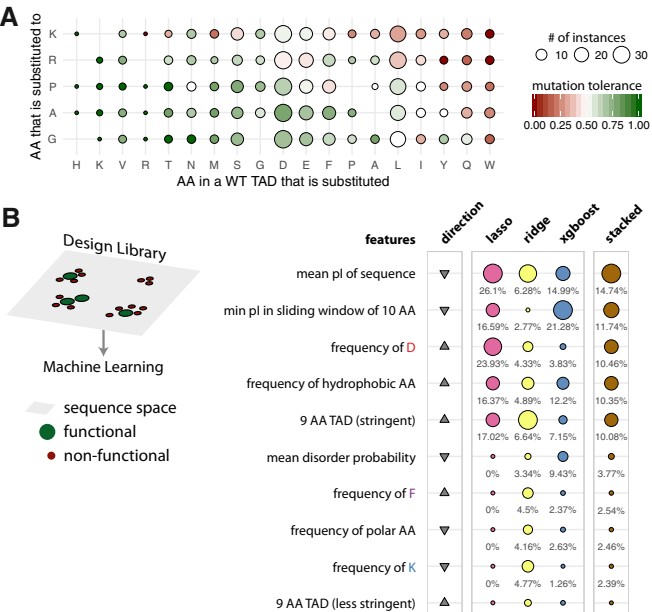

**Figure 4. Mutational scanning of naturally occurring TADs and the top 10 features of the machine-learning models trained on the design library.**

A  Heatmap of the tolerance to amino acid substitutions in WT transactivation domain sequences. The tolerance of a mutation is defined as the fraction of functional sequences over all the sequences when a specific substitution was performed. The columns (amino acid in a WT TAD that is substituted) are ordered according to decreasing tolerance (from left to right), and the rows (amino acid into which a residue in the WT TAD is substituted for) are ordered according to decreasing tolerance (from bottom to top). The cells are colored on a green to red gradient for high to low tolerance, respectively. Empty tiles represent data points not detected in the library.

B  Schematic describing the sequence space explored by the design library (left). Table listing the top 10 most important features. The relative feature importance is given as relative percentages in the last four columns. The size of the circles is scaled per method (lasso, ridge, xgboost, stacked). The direction column denotes the direction of enrichment of the given feature for functional sequences compared to non-functional sequences (up, positive direction and down, negative direction). This figure provides a simplified description of the actual features, which are available in Table EV7.

| features | direction | lasso | ridge | xgboost | stacked |
|---|---|---|---|---|---|
| mean pI of sequence | ▽ | 26.1% | 6.28% | 14.99% | 14.74% |
| min pI in sliding window of 10 AA | ▽ | 16.59% | 2.77% | 21.28% | 11.74% |
| frequency of D | ▲ | 23.93% | 4.33% | 3.83% | 10.46% |
| frequency of hydrophobic AA | ▲ | 16.37% | 4.89% | 12.2% | 10.35% |
| 9 AA TAD (stringent) | ▲ | 17.02% | 6.64% | 7.15% | 10.08% |
| mean disorder probability | ▲ | 0% | 3.34% | 9.43% | 3.77% |
| frequency of F | ▲ | 0% | 4.5% | 2.37% | 2.54% |
| frequency of polar AA | ▽ | 0% | 4.16% | 2.63% | 2.46% |
| frequency of K | ▽ | 0% | 4.77% | 1.26% | 2.39% |
| 9 AA TAD (less stringent) | ▲ | 0% | 3.54% | 2.27% | 2.1% |

known TADs from diverse organisms ranging from yeast to human that were functional in our HSF1-based selection assay. These include TADs from human KLF4, ESX, yeast Pdr1, Oaf1, plant HSFA2 and viral VP16 and EBNA2 and includes artificial TADs identified from previous studies (Table EV4). In addition, we created variants of these reference TAD sequences to investigate their functionality (~ 90 bp, < 30aa; 962 variants; design library; Table EV5). Guided by the observations from the random library, we performed mutational scanning of all positions of the reference TAD sequence to study the importance of a residue in the natural sequence and their ability to tolerate positive charge (K and R scanning), conformational changes (P and G scanning) as well as alanine (A scanning). Some variants in this library also include known single nucleotide polymorphisms in the human population and disease mutations (variants seen in cancer genomes) in the human TADs (Table EV6).

A detailed analysis of this mutagenesis data involving 962 sequences/variants revealed that the introduction of a positive charge instead of any residue in the reference TAD sequence was the least tolerated mutation (Materials and Methods). Introduction of G and A were the most tolerated mutations (Fig 4A; see rows) except if the reference TAD residue is a W, Q, I, L, and Y. Introduction of a P, which typically constrains the conformation of a peptide bond, is less tolerated by aromatic residues, whereas polar and negatively charged residues appear to tolerate P better. These findings suggest that aromatic and bulky hydrophobic residues are critical for naturally occurring TAD sequences and negatively charged residues can be substituted possibly due to their redundancy (D and E are among the most frequently occurring amino acids in wild-type TADs). These observations are in line with what we observed in terms of amino acid enrichments among the functional sequences in the random library.

Some of the sequence variants that represent polymorphisms in the natural human population and in cancer genomes do not confer survival in our assay for TAD functionality. For instance, the W30R in EKLF4 (allele frequency in the human population of $1.3 \times 10^{-5}$; gnomAD database) and the E135K mutation in ESX, which is prevalent in esophageal adenocarcinoma (allele frequency of 0.37 from cBioPortal), may lead to loss of TAD activity in these transcription factors (Appendix Fig S10). Thus, IDR-Screen can be a powerful framework to screen and infer the functional impact of a large number of naturally occurring SNPs and mutations observed in disease genomes.

We then investigated the individual TADs in terms of their ability to tolerate mutations. To this end, we computed the tolerance score for every TAD in our design library. Tolerance is defined as the number of variants that confer survival over all variants tested for that TAD. An analysis of the distribution of the tolerance scores of the TAD sequences revealed a unimodal distribution where a

majority of the TAD sequences have intermediate tolerance scores (Appendix Fig S11). VP16 and Oaf1 are at the most tolerant end of the spectrum, whereas Gln3 is in the least tolerant end of the spectrum. This suggests that most TADs are tolerant to mutations to a certain extent and this is likely to be determined by the nature of the substitution (Fig 4A). It also suggests that naturally occurring wild-type TADs emerged during evolution to be more or less tolerant to different kinds of mutational perturbations with implications for fine-tuning of function via sequence polymorphisms.

ML-based learning of the design library data using the different approaches and by a stacked model shows improvement in predictive capacity (for the best performing models: precision–recall AUC: 0.7972; random performance: 0.5626 and ROC-AUC: 0.7602; random performance: 0.5) and allowed the identification of the key features that are important in naturally occurring TADs (Fig 4B; Appendix Fig S12A and B; Table EV7). The top 10 important features include the 9-aa TAD motif, pI of the sequence, single amino acid (enriched: D, F; depleted: K) and grouped amino acid composition (enriched: hydrophobic; depleted: polar) and lower disorder probability score.

## Combining libraries can train models that are more general and guide design of new sequences

To develop a more general predictor and identify features that are important to discriminate sequences that have the potential to function as a TAD in our system, we combined the sequences from the random and design libraries to train machine-learning algorithms (Fig 5). Using this combined library, we trained models that strike a balance between not being able to pick up discernable pattern due to broad and sparse sequence space (random library) versus being biased by picking up patterns from a dense and narrow sequence space (design library; Appendix Fig S13A and B; for the best performing models: precision–recall AUC: 0.2001; random performance: 0.0206 and ROC-AUC: 0.7735; random performance: 0.5). An investigation of the features that contribute the most to prediction revealed several features such as the degenerate mini-motifs (also seen in the analysis of the random library) as well as the 9-aa TAD motif (also seen in the analysis of the design library) with variations. The most consistent feature that appears to be important in all three libraries is the pI of the sequence (Fig 5; Table EV8).

To explore this further, and considering that the degenerate mini-motifs emerges as an important feature to make a functional sequence, we generated new sequences and tested their ability to be functional in our assay. More specifically, we tested sequences that had a repeat of just aromatic and acidic residue and their combination thereof with the absence of a positively charged residue (i.e., Wx10, Dx10, and WDx10, respectively). The designed WDx10 sequence based on the findings presented here performs comparably to WT HSF as well as VP16 TAD in terms of conferring survival and growth rate (Fig 6).

## Discussion

Based on the findings described above and using the current mechanistic model, we present a general description of what constitutes a

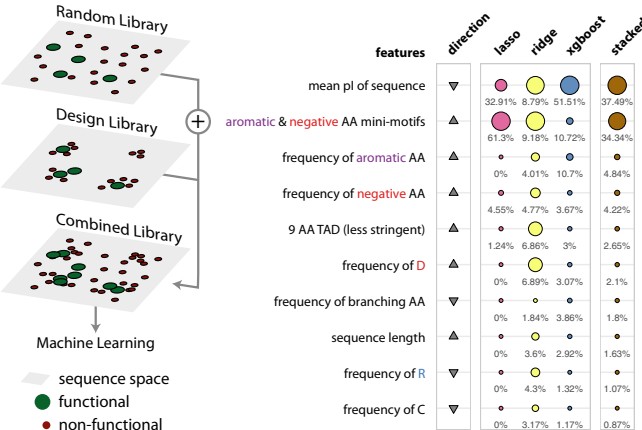

**Figure 5.    The top 10 most important features of the machine-learning models trained on the combined library.**
Schematic describing the sequence space explored by the combined library (left). Table listing the top 10 most important features. The relative feature importance is given as relative percentages in the last four columns. The size of the circles is scaled per method (lasso, ridge, xgboost, stacked). The direction column denotes the direction of enrichment of the given feature for functional sequences compared to non-functional sequences (up, positive direction and down, negative direction). This figure provides a simplified description of the actual features, which are available in Table EV8.

TAD (Fig 7). Enrichment for the negatively charged residues may ensure that the segment is in an extended unstructured conformation, repelled away from the DNA phosphate backbone to encounter the components of the transcriptional machinery that harbor positively charged patches. The aromatic and bulky hydrophobic residues within TADs can engage with the subunits of the transcriptional machinery through diverse modes of interactions (e.g., pi–pi, cation–pi, hydrophobic) and may bury within hydrophobic binding pockets. Hence, a combination of acidic and aromatic/hydrophobic residues with distinct spatial patterning along with the absence of positively charged residues provides a scenario of an extended disordered peptide that is "peppered" with anchoring residues, which can then interact with sufficient strength with appropriate protein interfaces. Given the degeneracy and redundancy of mini-motifs within TADs, the binding strength is unlikely to come from highly specific and strong interactions, but rather from multiple individual binding events (i.e., fuzzy complex, Tompa & Fuxreiter, 2008). The high degree of flexibility may permit the aromatic anchor residues to lock in the binding pocket or engage in stabilizing interactions on the surface of the components of transcriptional machineries. The discovered sequences might also facilitate transcription through alternative mechanisms such as disruption of promoter nucleosomes either by interacting with positively charged histones and hydrophobic crevices of nucleosome or intercalating between bases of nucleosomal DNA, thus triggering promoter chromatin remodeling (Erkina & Erkine, 2016; Fang et al, 2016). In this manner, TAD functionality may be an emergent property that depends on a combination of the context (amino acid composition) and spatial patterning (sequence motif) of particular amino acid types within disordered segments.

In this context, it is interesting to note that 1% of the random sequences were functional. This is a large number especially

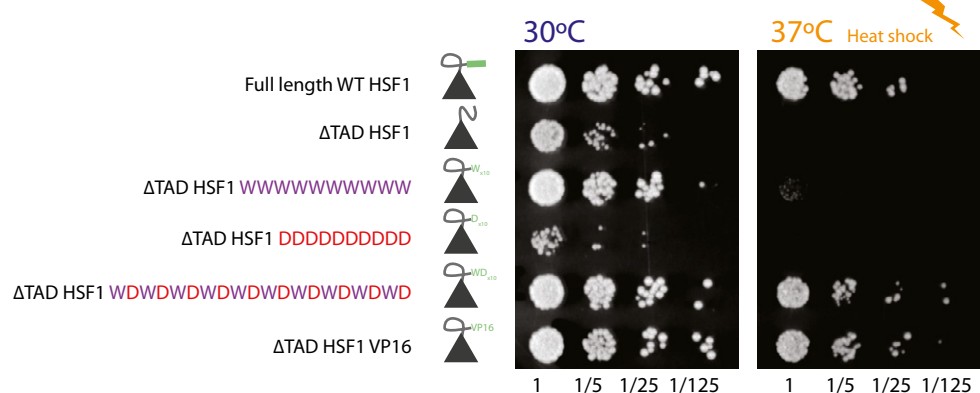

**Figure 6.   Spot-dilution assay of designed sequences.**
Spot assay of designed TAD constructs. The spot assay was performed at 30°C (left) and during heat shock at 37°C (right panel).

considering that a short and completely random sequence can decide between life and death of the organism hosting it. From the analysis of the structures of TADs in complex with co-factors (Appendix Fig S9), it is clear that different sequences adopt similar binding mode when interacting with the various co-factors. This suggests that a large number of sequences can and are compatible with co-factor interaction. Furthermore, many sequences can fulfill the role of a TAD by interacting with one of many components of the transcriptional machineries, and hence, a larger fraction of the sequences may confer survival during the selection experiment. From a technical perspective, the random peptide is selected for function and not binding to a specific protein domain. In other words, the sequences are not explicitly selected for binding to a specific component of transcriptional machineries but to any of the over 100 proteins that are involved in transcriptional initiation. These reasons could possibly explain why such large number of functional sequences could be detected.

We would like to emphasize that the specific design of the TAD assay is important for determining and interpreting the functional and non-functional sequences. Here, we used a survival-based selection assay to identify sequences that can launch the heat shock response at a wide range of promoters. One could also design other assays that can select sequences based on the expression level of a reporter (e.g., GFP). Furthermore, one could design more specific assays to obtain detailed mechanistic insights. For instance, the same libraries can be screened in different genetic backgrounds such as knockouts of specific mediator components to identify sequences

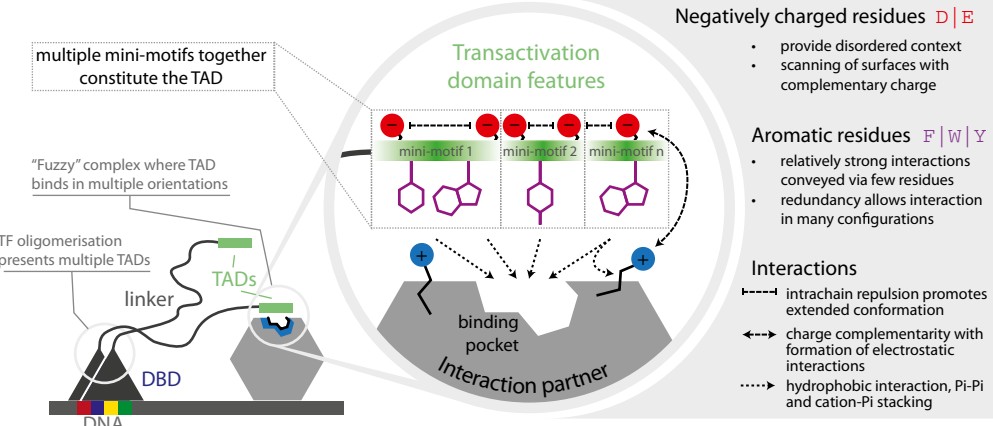

**Figure 7.   A mechanistic model for TAD function based on findings from this study.**
Transcription factors (TFs) interact with DNA via the DNA binding domain (DBD, blue triangle) and to their interaction partners (gray hexagon) via their transactivation domains (TAD, green rectangle). The enrichment for negatively charged residues leads to a local extended conformation of the TAD via intra-chain repulsions, providing the appropriate context for aromatic residues to be exposed and to bind to their interaction partners in hydrophobic binding pocket (circular inset). The aromatic residues could fit the pocket in a stochastic manner, binding in different configurations. This would result in the formation of a "fuzzy" complex. In this case, the negative charges could furthermore contribute to the affinity of binding given that the TAD-interaction surfaces often expose positively charged patches. The absence of positively charged residues, compositional bias, and particular spacing of negatively charged and aromatic residues could hence be considered as giving rise to a collection of short mini-motifs that collectively contributes to TAD functionality.

that depend on their ability to recruit the deleted factors. The sequences identified could be interpreted in the context of existing mechanisms of how transactivation domains perform their function as discussed above; hence, the assay used here is more general and less dependent on the specific mechanism (e.g., interaction with a particular component of the machinery).

Nevertheless, the functional sequences identified here constitute a good starting point to perform more focused follow-up studies such as identification of interaction partners through pull-down experiments to infer new mechanisms of action (e.g., interaction with previously uncharacterized subunits of the machinery) or their structural characterization. It is worth noting that some of the sequences that were non-functional in our assay may still be functional in other biological contexts because of the differences in specific aspects of the assay design such as the organismal setting, the differences in transcriptional machineries, the extent of negative design, nature of the promoter, DNA binding domain and the nature of the assay (e.g., survival versus reporter expression). Thus, with different types of assays and by using larger, and different libraries of sequences, the predictive models will become better and our understanding of what constitutes a functional TAD can be much more precisely defined. In this context, including and developing novel features based on new understanding that quantitatively describes sequences can allow better characterization of what makes a functional TAD.

In conclusion, the IDR-Screen framework described here allows for the discovery and characterization of biologically and functionally relevant disordered regions as well as enhances our understanding of what features make such sequences functional. Here, we have applied IDR-Screen to study transactivation domains. The approach presented can be readily expanded to study other functions that are mediated by disordered regions such as protein degradation. Employing this framework iteratively could lead to the development of models that can provide insights into sequences that have not yet been experimentally investigated with increasing reliability. Such predictors can then be used to scan protein sequences, alternatively spliced protein regions, and entire proteomes of organisms to discover potential functional disordered sequences and/or to assess their potential impact on functionality upon perturbing sequences (e.g., mutations that are seen in cancer genomes and natural variation). In this manner, we anticipate that adapting IDR-Screen can help uncover the function of disordered regions from any organism and of the dark proteome (Kruger, 2016) as well as to design new sequences with defined functional properties.

# Materials and Methods

### Library construction and screening

The parental library cloning vector pTYE3000 was constructed by insertion of a truncated version of yeast HSF1 (positions −1,018 to +1,479 encoding HSF amino acids 1–493) into a poly-linker site of pRS314 (http://www.snapgene.com/resources/plasmid_files/yeast_plasmids/pRS314/). This was done using synthetic restriction sites KpnI and NotI that were introduced during PCR amplification of the indicated HSF1 fragment. The sequences of the PCR primers are provided at the end of this section. In addition, a unique AscI site was engineered at the 5′-end of HSF1 truncation. The plasmid

construct was confirmed by sequencing. The AscI and SacI sites of the vector (SacI is 18 nucleotides downstream of NotI) were used to insert a synthesized 60-nucleotide random DNA oligo library containing three stop codons in all alternating frames at the 3′ end. The design DNA library containing individual wild-type TAD sequences and their variants was synthesized by CustomArrays, amplified according to the manufacturer instructions using PCR primers containing restriction sites for cloning and ligated into the parental library vector. After transformation into DH5-alpha supercompetent cells, each library (random and design) contained ≥ 300,000 independent clones with at least 80% containing inserts. Insertions, retention of the reading frame, and the cloning sites were confirmed by sequencing. The individual plasmid libraries were isolated from bacteria and transformed into PS128 (MATa; ade2–1; trp1–1; can1–100; leu2,3–112; his3–11; ura3; hsf1::LEU2), selecting for Trp+ prototrophs. The transformation plates were replica-plated onto –trp 5-FOA containing plates, half of which were incubated at 30°C and the other at 37°C for 0, 2, 4, 6, and 8 days. Cell samples were collected at each time point, total DNA was isolated from each sample, and the library inserts were PCR amplified using primers containing Illumina adaptor and barcode sequences. Next-generation sequencing (NGS) of the resulting pools was done using Illumina MiSeq platform.

Forward primer: GATCAGGTACCATTCATGCTTACGATAAAATCAC TTGGA.
Overhang: GATCA; Kpn1 site: GGTACC; wt HSF1: ATTCATGCTTAC GATAAAATCACTTGGA

Reverse primer: CGGTGGCGGCCGCTTACTATAGATCTGGCGCGCC TATATTTTTCTTTCTGTT GGTGGTATTAG
Overhang: CGGTG; Not1 site: GCGGCCGC; Insert (replaced): TTAC TATAGATCT;
AscI site: GGCGCGCC; wtHSF1: CGGTGGCGGCCGCTTACTATAG ATCTGGCGCGCCTATATT TTTCTTTCTGTTGGTGGTATTAG

### Collection of wild-type TADs and their variants

A set of known wild-type TADs were identified from the literature and were selected to cover a diverse range of organism including yeast, virus, human, and plants. They span an amino acid length range from 8 to 21 with an average of 16 (see Table EV4). This set of sequences was used to design a library of sequences with different mutations. A first set of variants consists of a point mutation at every position of the wild-type TAD with either of the following amino acids (A, G, P, K, or R). The set of resulting sequences includes disease mutations as well as natural variation that were collected from COSMIC (http://cancer.sanger.ac.uk/cosmic), cBioPortal (http://www.cbioportal.org/), and gnomAD (http://gnomad.broadinstitute.org/), respectively (see Table EV6).

### Read preprocessing to obtain variant counts

The sequencing read processing stage resembles metagenomics pipelines [e.g., UPARSE (Edgar, 2013) or `vsearch` (preprint: Rognes *et al*, 2016)] with adjustments to parameters as described below. The sequences of the reads from the different samples (i.e., time points) were first combined making sure that their sample of

origin information was kept in the fastq headers of each read. The forward and reverse reads were then merged using `vsearch` in order to produce higher quality resulting reads. Only reads that would merge into sequence of length 60 bases were kept (the length of the input DNA), while allowing for a maximum of two gaps. Sequences were then filtered using `vsearch` at an expected error rate of 1, i.e., not tolerating more than 1 expected error across the read after taking the quality scores per base into account. Next, the replicate barcode sequences (present in the design library) were removed using `cutadapt` (Martin, 2011), recording the replicate barcode identifier in the fasta header. Barcodes were designed to have a length of eight nucleotides and to have a Levenshtein distance of at least four, i.e., the distance that consists in the number of substitutions, insertions and deletions required to change one sequence to another. Barcodes were assigned after accepting an error rate of 25% just within the barcode sequence (not the entire length of the read). For a barcode of length of eight nucleotides, this translates to a maximum of two mistakes (0.25 * 8 = 2). This stringent design allowed us to accurately distinguish the reads that come from the different replicates or samples. Finally, the remaining adaptor sequences were removed using `cutadapt` to produce the final set of raw sequences of the reads that are to be quantified for their relative abundance, again allowing for an error rate of 10% only in the adaptor sequences (length of 21 nucleotides).

There are two related protocols depending on the origin of the library. For the random library, the reads are dereplicated using `vsearch`; i.e., only unique sequences are kept, and their count is recorded, respectively. Only sequences with a count of at least two were kept in this set. Then, the dereplicated sequences were clustered into groups according to a minimum percent sequence identity of 90% using `vsearch`. This approach ensures that reads with minor sequencing errors (mismatches, insertions, or deletions) are not incorrectly identified as distinct sequences. Instead, they are clustered together to obtain a single sequence. Given the vast sequence space in a random 60mer DNA library, it is very unlikely that two distinct functional sequences from the experiment would be incorrectly "clustered" by this approach (Edgar, 2013). These groups (or clusters) are then considered a collection of centroids (each supported by a number of sequencing reads) that are used as the reference to realign the complete set of sequence reads to the closest centroid sequence using `vsearch`. The reads of the design library were directly mapped against the set of sequences that were designed (which are the centroids) again using `vsearch`. The sequence identity parameter was set to 80%. The number of reads mapped to centroid sequences for each sample was recorded (random library: three replicates—the replicate experiments; design library: six replicates—single experiment where each sequence was tagged with different set of barcodes that was used to separate the reads into the relevant samples in the post-sequencing data processing step above).

### Estimation of growth scores for each sequence

The raw sequencing counts for both random and design libraries were further processed to estimate growth rates of the different strains they are associated with. First, sequences with lower than three reads over the different time points were removed. Sequences with a count of 0 at the first time point (but nonzero counts at later

time points) were set to 1 in order prevent division by zero during the normalization step that follows (see below). Then, samples were normalized on a per-sample, per-replicate basis using the total sum normalization procedure, where every count is divided by the sum of all reads in the sample (per-replicate). The normalized counts were further normalized to the starting count. Next, we calculated the growth rates of individual sequences based on the normalized read counts. For each sequence, the normalized counts at different time points and their corresponding replicates are fitted with a robust linear model where the slope represents the growth estimate over time (intercept set to 1). Finally, the sequences were filtered based on the regression results. Sequences were retained if they had at least two replicates, at least 10 reads in total, and if the robust linear regression had converged. From all the sequences in the random library that were detected, ~ 67,263 fulfilled the stringent filtering criteria (see Table EV1).

### Processing growth scores to identify functional and non-functional sequences

To assess whether a sequence should be considered functional, we defined a growth estimate threshold. Above this threshold, sequences were classified as functional and below this threshold, sequences were classified as non-functional. For both the random and the design libraries, this threshold is based on the growth estimate for sequences that start with a stop codon; sequences starting with a stop codon should not be functional and therefore should represent the lowest possible growth estimate value. In both the libraries, there are sequences with stop codons at the start that represent truncated transcription factors. Therefore, the growth estimates of these sequences could be used to determine the split between sequences that are functional, even if the function is weak, and those that are not functional.

For the random library, growth estimate values were first rounded to two significant figures, reflective of the resolution of the experiment (values beyond two decimals are likely to only reflect noise in the regression procedure). The sequences were then classified into two groups: those where the insert started with a stop codon (3,139 sequences) and those that did not start with a stop codon (64,124 sequences). The former group is not expected to promote any survival (Appendix Fig S1B) at high temperatures and can therefore serve as the group for negative control (i.e., stop codon group). A threshold of growth estimate value was picked to achieve a balance between reducing the number of false-positive sequences at the same time as maintaining a large number of functional sequences for subsequent analyses. At a growth estimate threshold of −0.13, we would classify 19 of the 3,139 stop codon sequences as functional. This gives an FPR of 0.6% and allowed classification of ~ 739 sequences as functional (see Table EV1). Although this is an objective criterion to estimate the FPR, some sequences that are not expected to be functional (e.g., very short sequences) are classified as functional using the growth rate cutoff for a particular FPR. To better reflect the experimental false positives, we also considered all sequences with four or fewer amino acids as non-functional to obtain the percentage of false positives as 12.18% (90 sequences with four or fewer amino acids/739 sequences).

For the design library, three different stop codon sequences (i.e., variants) were included to help define the threshold growth estimate

value between functional and non-functional sequences (TAA with estimate of −0.21, TGA with estimate of −0.25 and TAG with estimate of −0.15). The highest growth estimate (−0.15) of the three stop codon variants was chosen to classify sequences into functional and non-functional groups. Anything above or below this value was classified as a functional or non-functional sequences.

It is expected that positive slopes of normalized read count indicate that the individual with that sequence grows over time, whereas negative slopes indicate that the relative fraction of the variant is decreasing. For example, if a strain with a sequence grows moderately over time, its fraction of total reads between subsequent time points might decrease because sequences that confer increased fitness are more likely to be sequenced in subsequent time points. In this way, when fitting the robust linear regression of the normalized read count over different sample collection time points, the read count might decrease, despite the sequence being moderately functional. Thus, the negative control of stop codon containing sequences provides a robust way to split the sequences in an objective manner.

**Converting sequences to vectors of features**

The feature calculation was performed with a custom software package in Python that can process large amounts of sequences and is extensible by other researchers and interfaces with existing software so that a broad range of features can be calculated. Each sequence is processed in three steps. In the first step, a numerical encoding (e.g., amino acid scale and disorder propensity) is defined to convert the sequences of amino acids in to a sequence of *numbers*. In the second step, different *views* (i.e., sub-sequences) of the full sequence are to taken to obtain various sub-sequences (e.g., sliding windows of defined sizes). Finally, the values for each *view* are *aggregated* (e.g., count, sum, mean, and max) to yield a final value. A whole range of combinations of these steps was applied to yield a total of 146 different features (see Table EV2; Appendix Fig S2). The features that were calculated this way were grouped into the following feature sets for interpretability:

1  The single amino acid composition (IVLAFWYGSPKRH-DETQNCM, respectively)
2  Grouped amino acid composition (Betts & Russell, 2003; Pommie *et al*, 2004; aliphatic: IVLA, aromatic: FWY, branching: VIT, charged: KRHDE, negative: DE, phosphorylatable residues: STY, polar: RKDEQNY, hydrophobic: VILFWCM, positive: KRH, sulfur-containing: MC and tiny: GASP)
3  The presence or absence of degenerate mini-motifs that are regular expression describing logical operation of pairwise combinations of fine and coarse amino acid groups (fine group: negative, positive, aromatic; coarse group: polar, non-polar). The amino acids can be interspersed by other residues with a spacing of 0–10 residues. The groupings were guided by amino acid enrichments of the functional sequences from the random library as well as other grouped amino acids (Appendix Fig S7).
4  Presence or absence of the 9-amino acid TAD motif (Piskacek *et al*, 2007; four different regular expressions: http://www.med.muni.cz/9aaTAD/)
5  Disorder propensity of the sequence (calculated using IUPRED, Dosztanyi *et al*, 2005)

6  The helicity of the sequence [as calculated by Agadir (Lacroix *et al*, 1998) and Heliquest (Gautier *et al*, 2008)]
7  General sequence properties which include molecular weight of the sequence (calculated using the BioPython package: biopython.org) as well as the sequence length (number of amino acids)
8  Isoelectric point, pI (calculated using the BioPython package: biopython.org)

**Machine learning on the sequences and their features**

The caret package in R was used to perform these tasks in combination with custom scripts.

*Preparation of the datasets*

First, the complete dataset with all the sequences, features and survival classifications was split into two subsets based on the library of origin, i.e., the random and design libraries. Furthermore, we constructed an additional dataset where the sequences from the different libraries were combined, i.e., the combined library. To train and evaluate models with the different datasets, we applied similar approaches for the analysis of each of the libraries with variations described below:

- *Random library dataset*: The dataset was first split into 75% for training (training set) and the remaining 25% for testing (testing set) to have a final testing set that simulates collection of new data in a realistic way. The entire machine-learning process was performed with $n$-repeated $k$-fold cross-validation ($k$ = 5 and $n$ = 10 repeats) on the training set (75% of original dataset). Thus, for every $k^{th}$ fold during cross-validation, the training set was split up into a training set of size (4/5$^{th}$) that was used to train the model, and a validation set (1/5$^{th}$) that was used to evaluate that model performance. This process was repeated 10 times (to ensure that the random splits into folds did not bias the performance). This yields a total of 50 data splits. Because of the unbalanced nature of the dataset, with fewer functional sequences (minority class), the dataset was subsampled to the minority class while training the model on the training set. To ensure we do not over-estimate performance, sub-sampling was not done on the validation or testing set (see Appendix Fig S3).
- *Design library dataset*: For the analysis of the design library dataset, a slightly different strategy was used. Specifically, we used group k-fold cross-validation ($k$ = 5) rather than k-fold cross-validation. The dataset is treated as 13 distinct groups of TAD sets, where each group is defined as a wild-type TAD sequence and all their variants. Therefore, for each data split, 4/5$^{th}$ of the 13 TAD sets (~ 10 TAD sets) were assigned to the training set and 1/5$^{th}$ (~ 3 TAD sets) were assigned as the validation set. This ensured that variants within a TAD set (which are highly similar) were not split between the training and the validation sets. This procedure was carried out 10 times, again yielding a total of 50 data splits. Note that because of the reduced dataset we did not split the design library dataset into an original training and testing set (75–25%) as for the random library. Because of the unbalanced nature of the dataset, the training set was subsampled to the minority class. This was not done on the validation sets (see Appendix Fig S3).

- *Combined library dataset*: We used the testing set of the random library as the testing set for the combined library. The different splits of the training set of the random and design libraries were combined to obtain the training set of the combined library. Similarly, the different splits of the validation set of the random and design libraries were combined to obtain the validation set of the combined library. The training set was subsampled to account for the library of origin and the unbalanced nature of the datasets in terms of the number of functional versus non-functional sequences. This procedure was repeated for each fold times the number of iterations (50 in total; see Appendix Fig S3).

### Feature preprocessing

High correlation between features for the sequence from the different library datasets was identified by computing a correlation matrix between all features using the Pearson correlation coefficient metric. The correlation matrix was converted into a network where features (nodes) would be connected if they were correlated (threshold of correlation coefficient of 0.75). From the resulting sub-networks (clusters) of correlated features, only one representative feature was kept (the feature least correlated to other features in the cluster). Unconnected sub-graphs were also retained. Different values for correlation thresholds were tested without significant differences to the outcomes of the modeling. Then, the dataset was checked for linear combinations of the features. The features were represented as a matrix, and QR decomposition (decomposition of a matrix into a product of an orthogonal matrix and an upper right triangular matrix) was applied to determine whether it was of full rank and to identify columns with dependencies. This approach was applied iteratively, removing columns with dependencies and checking the rank until no more dependencies were identified. From the feature sets with dependencies, one representative feature was kept (chosen at random), and the rest was removed. To allow interpretation and comparison to previous work, features previously associated with TADs, such as minimum sequence length, isoelectric point, hydrophobicity, disorder propensity, helicity, and the presence of the 9-aa TAD, were always included in the analysis, as well as at least one feature from each of the eight feature set described above.

### Model building

The different datasets were then used to train and evaluate models. For each of the iterations of the cross-validation (described above), prior to the model training, the following preprocessing steps were applied: (i) the data were centered and scaled; (ii) near-zero variance features, i.e., features that predominantly have the same value for all objects were removed at a ratio cutoff of 95–5 % for the most common value to the second most common value (default value in caret package; https://CRAN.R-project.org/package = caret).

Simple logistic regressions (logit) were applied to estimate the predictive power of previously reported sequence features individually (Appendix Fig S5A) and were combined together in a model that considers all these features (Appendix Fig S6A and B). Separate logistic regressions were fitted for length, pI, hydrophobicity, disorder, helicity and presence of 9-aa TAD. The logistic regression builds a model that indicates the probability to be a functional or non-functional sequence at a given value of the different properties, respectively.

When dealing with multiple features, Lasso and Ridge (James *et al*, 2013; from "glmnet" R package) and xgboost (Chen & Guestrin, 2016; from "xgboost" R package) algorithms were applied because of their intrinsic feature selection routines, while differing in their nature to identify statistical patterns in the data that can have very different underlying generative phenomena. Lasso (least absolute shrinkage and selection operator) is a regression method that penalizes the absolute size of regression coefficients (James *et al*, 2013). Varying the degree of penalization constrains the estimates to become zero (i.e., removes them from the model), which helps (i) to deal with features that have a high degree of correlation (i.e., similar features are filtered out) and (ii) to describe feature importance (i.e., intrinsic feature selection) as features that are important will require very high penalties not to be included in the model. We used the penalization on coefficients of logistic regression models. Ridge regression works in a similar way, penalizing the inclusion of additional features, but generally includes more features, albeit with lower coefficients (James *et al*, 2013). Boosted trees (xgboost) work on a very different premise and are based on ensembles of shallow decision trees that are built in a stagewise fashion focusing on data points that are hard to classify correctly (boosting). It produces an ensemble of weak learners that together give accurate predictions. These models also provide a way to assess feature importance that is based on the reoccurring placement of features high up in the ensemble of decision trees. Used in conjunction, the two very different approaches with intrinsic estimation of feature importance provide a way to learn the reoccurring patterns in the dataset and interpret them more confidently.

For the lasso and ridge models, hyper-parameter tuning was applied with a grid search over lambda values from 0.00001 to 0.1 over 50 evenly interspersed values (with alpha fixed to 1 and 0, respectively). The parameter space explored for the xgboost models was eta = 0.1 or 0.3, max tree depth = 2 or 4, features used by each tree were set to 0.8, sub-sampling of sequences was set to 0.5 and 0.75 (i.e., half the sequences or three quarters), the number of rounds was set to 30 or 100, a gamma value of 0 was applied, and the minimum child weight was set to either 0.5 or 1. All possible combinations of these hyper-parameter values were tested.

Stacked models (or meta-models) were constructed from the three (Lasso, Ridge, and xgboost) best primary models (or base models) using the "caretEnsemble" package. This combines the best base models using a linear regression based approach. The resulting ensemble classifiers use the predictions (probability values of being functional) of the different validation sets (described above), which were used to evaluate the different base models. To estimate performance of the ensemble model, a bootstrap sampling procedure is applied ($n = 50$). For each bootstrap iteration (with replacement), the dataset was split into training and validation sets, subsample the training set to the minority class, train the model, and evaluate performance on the validation set. This is used to obtain weights for the different base models in order to obtain the final ensemble model.

*Evaluating model performance*

Model predictions during hyper-parameter tuning (grid search) as well as the final model evaluations on the random library testing set were evaluated with the precision–recall AUC evaluation metric. A model was constructed for each parameter set, and it was trained and tested on multiple dataset splits (as described above). For each split the evaluation metrics were calculated. To select the best final model, the simplest model that was one standard error from the best model was selected to prevent over-fitting that can occur in the best model. The ordering of models from simplest to most complex was done according to the caret package model specifications. For boosted trees, lowest number of iterations, followed by most shallow tree depth was used as the criterion. For the penalized generalized logistic regressions, they were ordered first on the L1 penalty followed by the L2 penalty. True-positive rate (TPR) is defined as the ratio of true positives over the sum of true positives and false negatives (all positives). False-positive rate (FPR) is defined as the ratio of false positives over the sum of false positives and true negatives (all negatives). FPR is also expressed as 1-specificity where specificity is ratio of true negatives over all negatives.

*Structures of TAD interactions*

The structures of multiple complexes of transactivation domains (TADs) binding their respective partners were collected from the PDB whose codes are provided within parentheses: Tfb1-VP16 (2K2U), NcoA–1-STAT6 (1OJ5), MDM2-p53 (1YCQ), CBP-CREB (1KDX), TFIIB-VP16 (2PHG), PC4-VP16 (2PHE), Gal11-Gcn4 (2LPB), and Med25-VP16 (2KY6). Structures of the complexes were aligned based on the helical region of the respective TAD that they contain, centered on the central hydrophobic residue of the helix. The surfaces on the interaction partners were colored according to their electrostatic potential (blue: positively charged, red: negatively charged in Appendix Fig S9).

*Spot-dilution assay*

Yeast cell cultures expressing ΔTAD HSF1 fused to a corresponding TAD sequence were grown to mid-log phase in the YPDA medium. An 1 ml aliquot was taken from each culture and washed twice with sterile water by centrifuging samples for 5 min at $1,000 \times g$. Washed cells were re-suspended in sterile water to an OD600 = 0.3. A fivefold or threefold (see figure labeling) serial dilution for each culture was prepared in sterile 96-well plates. Using a replica-plater, the diluted cultures were transferred to Petri dishes with solid media. Replica plates were incubated at 30°C or 37°C. Images were taken after 42 h of growth, and the same imaging settings were used for all captures. Strains carrying specific plasmids were either individually constructed or randomly selected from the pool at the colony growth stage. The sequence of all constructs was confirmed by individual DNA sequencing.

**Data availability**

The TAD predictor is available online at https://github.com/cn jr2/tad-prediction/. Datasets are available as Tables EV1–EV8.

**Expanded View** for this article is available online.

## Acknowledgements

We thank E. Chen, B. Broyles, and N. S. Latysheva for help with ML applications and conceptual discussions. We thank I. Huppertz and Y. Sugimito for their help on next-generation sequencing and Jenna Hebbe for assistance with yeast spot-dilution assay experiments. We also thank S. Chavali, P. Lakshminarasimhan, R. Pancsa, X.-H. Li, and B. Porebski for their comments on this work. This work was supported by the European Research Council Consolidator Grant (IdrSeq; 682414; M.M.B., C.N.R.J.), Medical Research Council (MC_U105185859; M.M.B. and C.N.J.R.), the AFR scholarship from the Luxembourg National Research Fund (C.N.J.R.), the Marie Curie Fellowship (G.D.B), the National Science Foundation (MCB-1029254; A.M.E. and T.Y.E), and the Holcomb Award and Innovation grant from Butler University (A.M.E). M.M.B. is a Lister Institute Research Prize Fellow.

## Author contributions

CNJR, AME, and MMB designed the project. CNJR developed the experimental and computational design. AME and TYE performed the experimental work. CNJR and GDB performed sequencing experiments, collected, and processed the sequencing data. CNJR wrote the scripts and performed all the analysis with GDB and DCD providing inputs on feature engineering. CNJR and MMB analyzed the results and wrote the manuscript with inputs from all authors. AME and MMB supervised the project.

## Conflict of interest

The authors declare that they have no conflict of interest.

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
