## [Review Process File · Molecular Systems Biology]

High-throughput discovery of functional disordered regions: investigation of transactivation domains

Charles N. J. Ravarani, Tamara Y. Erkina, Greet De Baets, Daniel C. Dudman, Alexandre M. Erkiné & M. Madan Babu

Review timeline:

Submission date:	8 January 2018
Editorial Decision:	7 February 2018
Revision received:	27 March 2018
Editorial Decision:	3 April 2018
Revision received:	10 April 2018
Accepted:	11 April 2018

Editor: Maria Polychonidou

Transaction Report:

1st Editorial Decision

7 February 2018

Thank you again for submitting your work to Molecular Systems Biology. We have now heard back from two of the three referees who agreed to evaluate your study. We have not yet heard from reviewer #3 but since the recommendations of the other two referees are quite similar I prefer to make a decision now rather than further delaying the process. The two reviewers are overall supportive. They raise however a series of concerns, which we would ask you to address in a revision of the manuscript. If we receive comments from reviewer #3 within the next few days, I will forward them to you so that you can address them in your revision.

The reviewers' recommendations are rather clear so I think that there is no need to repeat the points listed below. Please let me know in case you would like to discuss any of the points in further detail.

REVIEWER REPORTS

Reviewer #1:

In this short manuscript the authors set out and provide demonstration of a general strategy to map out the sequence requirements for functioning of intrinsically disordered protein domains. The method describes screening of a random or designed library of peptides linked to some functional in vivo assay, clonal selection by coupling survival to some enzyme activity or fluorescent reporter, next generation sequencing of positive clones and finally, a machine learning method to tease out sequence features and properties that dominate positive selection.

The strategy was applied in the yeast *S. cerevisiae* in which survival-selection was determined by expression of a selection gene product under the control of a promoter controlled by a transcription factor in which its activation domain has been replaced with random or directed libraries coding for 20 amino acid peptides.

Subsequent machine learning on resultant positive clones revealed several features of interest, both predicted (selection for negatively charged amino acids) and not so very obvious very strong selection for tryptophan in the sequences of the artificial activation domains. Tests with directed library revealed a novel DW tandem repeat as providing maximum activity. To the best of my knowledge, such a sequence has never been demonstrated to regulate transcription with such high activity.

Overall, the study is well performed and the manuscript is well written. The screening method itself is not particularly novel in comparison to many other screening strategies. It appears to me, however, that the combination of the screen with data analysis could provide more insight into binding of IDPs to other molecules than more conventional analyses.

I return again to the DW repeats. These are interesting for two reasons: First, while in vitro binding screens such as phage display have been shown to exhibit unusual amino acid biases, notably for abundances of tryptophan, I am not aware that in vivo screens show the same bias. This issue is discussed and the structural and thermodynamic consequences of multiple Trp tandem repeats were discussed and beautifully illustrated in the strange case of the "Trp Zipper", an extraordinarily stable beta hairpin discovered in a phage display screen (Cochran, A, et al. PNAS, 2001). Another unrelated structure that would be worth discussing is the co-crystal structure of the messenger RNA 5' cap-binding protein (eIF4E) bound to 7-methyl-GDP (Marcotrigiano, et al. Cell, 1997). Here the 7-methyl-GDP is nestled between two tryptophan rings forming an interesting and subtle interaction that might reflect something about the way that the author's weird TD repeat interacts.

So all in all this is an interesting study that should be of interest to the general readership of MSB. I must say, however, that before it is accepted I'd suggest major revision to the figures, notably figure 1, 3a, b, d and 4a, b. These are not of the quality, elegance and clarity that I have come to expect from this group and make difficult to follow, what should be straightforward ideas. Figures 3b, d and 4b are the worst and need to be substantially simplified and clarified to provide clear information that a general reader can appreciate.

Reviewer #2:

This manuscript describes a high-throughput method for finding what properties make a sequence activate transcription of a reporter gene when fused to a DNA-binding domain binding to the promoter of the reporter gene. Only a good handful of so-called activation domains are known from low-throughput experiments. This work represents so far the first high-throughput, unbiased screen of activatory peptides.

Although a relatively low number of only 760 activating sequences were found in the screen, the authors were able to demonstrate in a cross-validation test that a machine learning classifier trained on a part of the activating and non-activating sequences can distinguish new activating and non-activating sequences (at 1:1 ratio) with a precision much higher than random expectation. The manuscript therefore makes a unique and valuable contribution to the very challenging topic of transcriptional activation.

Major points:

1. According to the description in the Methods section, it seems that sequencing error rates are very high, because an error rate of 25% in the barcode are still accepted: 1 out of 4 nucleotides can be misread. That is an extremely high error rate and it would be important to comment on the causes and to discuss more the possible consequences for interpretation of the data.
2. The authors describe that they cluster the sequences into clusters of similar sequences with a

minimum sequence similarity threshold of 90%. It is not clear a priori why this should be necessary, as one would expect that in a random library sequences with different barcodes would be unique, and the chance of two sequences of length 15 residues to be the same by chance is on the order of $20^{-(15)} = 0$. The authors need to discuss the possible origin of these similar sequences.

Also, when the authors write that "The resulting mapping table for both the random and the designed libraries was aggregated to a count table where every sequence has its count recorded for each replicate", do they mean that every ****centroid**** has its counts recorded, which are the aggregate of all members of its cluster of similar sequences? Does it mean that 760 activating ****centroid**** sequences were discovered, or does the number 760 include the very similar sequences?

3. The authors need to make sure that in their cross-validation benchmark on the machine learning predictors, none of the training sequences has a similarity to any of the test sequences higher than expected by chance. Otherwise, the prediction performance could be highly overestimated, as test sequences for which similar sequences have been trained on will be predicted much more accurately than the realistic cases in which no similar training sequence has been observed.

The authors mention that many of the sequences they obtain are very similar to each other, hence the clustering. But how many are similar but have sequence identities below 80% or 90%? If pairs with sequence identities higher than what can be expected by chance (10-20%) are split between training and test set, the estimated performances of the predictors will be overestimated. A simple criterion could be to limit the sequence identity in a local alignment of the coding sequences to 30%. Here, sequence identity is defined as the fraction of identical aligned residue pairs to the length of the shorter sequence.

4. There is a contradiction on page 18 in the online methods: "The dataset was split into 75% for training and the remaining 25% for testing..." This implies that the authors used 4-fold cross-validation, but in the next sentence they write "The entire machine learning process was performed with repeated cross-validation (k = 5 folds and 10 repeats) on the training set." The entire procedure of cross-validation and sampling is not clear, as the authors do not seem to have used the standard cross-validating procedure. For example what does "10 repeats" refer to?

Minor points:

5. Instead of demonstrating the usefulness of the learned features of activating domains using the sequence WDWDWDWD..., it would be more convincing to show the activation potential for a less obvious sequence where the authors make use of the specific minipatterns they have learned. The prediction of WDWDWDWD... could have been made even before because it is known from earlier work using site-directed mutagenesis (<http://www.pnas.org/content/111/34/E3506.full.pdf?withds=yes>) that mutations to W generally increase activation potential and that negative charges are important to prevent aggregation.

6. The description on top of page 17 seems to mean that the training ****and**** test sequences were subsampled to get a ratio of 1:1 between activatory and non-activatory sequences. However, the Supplementary Figures 6 and 9 show a dotted line with an "imbalance ratio" of 0.0116. This imbalance ratio should be 0.5, shouldn't it? On the other hand, if test sequences were ****not**** subsampled, I would expect an imbalance ratio of $760 / 67000 = 0.113$ and not 0.116.

7. Figures S6 and S9: The x axis labels "1 - sensitivity" should be "1 - specificity". The FPR needs to be defined explicitly in the online methods.

8. In the main text (Page 3) the authors write "...we obtained robust measurements for 67,000 variants." In the methods, the authors write "Given the imbalance in our dataset (760 functional: 65,517 non-functional...)". This adds up to 66270, not 67000. If the authors want to round, it should be 66000.

Note to all reviewers

We are pleased to note that all the referees feel we have developed a useful method that allows to screen for functional regions in intrinsically disordered regions, and that we could successfully apply it to the challenging topic of transcription activation that is of interest to the scientific community:

Reviewer #1: ... *“Overall, the study is well performed and the manuscript is well written. The screening method itself is not particularly novel in comparison to many other screening strategies. It appears to me, however, that the combination of the screen with data analysis could provide more insight into binding of IDPs to other molecules than more conventional analyses.” ... “So all in all this is an interesting study that should be of interest to the general readership of MSB.” ...*

Reviewer #2: ... *“Only a good handful of so-called activation domains are known from low-throughput experiments. This work represents so far the first high-throughput, unbiased screen of activatory peptides.” ... “The manuscript therefore makes a unique and valuable contribution to the very challenging topic of transcriptional activation.”*

Reviewer #3: ... *“This work is the first attempt to examine functionality of non-motif disordered regions (as far as I am aware) using a large-scale approach and should thus be of interest to a large audience. Their main result (that a large fraction of sequences can work as TADs) is interesting and novel.” ... “In summary, this is a very interesting paper in an important field that neatly combined high throughput methods with modern analysis methods” ...*

Given the generally enthusiastic comments from the referees, and the highly constructive criticisms raised by them, we would like to suitably revise the paper for further consideration at *Molecular Systems Biology* as an article. In this document, we provide a point-by-point response to the referees' comments along with the action taken for the revised paper.

To help the reviewers and the editor go through our point-by-point responses, we have the reviewers' statements in *italics* and our responses in normal text in blue. For each comment, we provide a suggested action that we propose to undertake while preparing a revised version of the paper wherein we also provide page numbers and line numbers in the revised manuscript.

Response to Reviewer #1: pages 2-3

Response to Reviewer #2: pages 4-9

Response to Reviewer #3: pages 10-13

We believe that by addressing the constructive criticisms raised by the expert referees in a revised manuscript, a considerably stronger paper has been produced. We therefore sincerely hope that the referees would support further consideration of a revised manuscript that addresses all the concerns.

Sincerely,

Charles, Alex and Madan

Point-by-point response to referees comments

Reviewer #1:

Comment 1.1: *In this short manuscript the authors set out and provide demonstration of a general strategy to map out the sequence requirements for functioning of intrinsically disordered protein domains. The method describes screening of a random or designed library of peptides linked to some functional in vivo assay, clonal selection by coupling survival to some enzyme activity or fluorescent reporter, next generation sequencing of positive clones and finally, a machine learning method to tease out sequence features and properties that dominate positive selection.*

*The strategy was applied in the yeast *S. cerevisiae* in which survival-selection was determined by expression of a selection gene product under the control of a promoter controlled by a transcription factor in which its activation domain has been replaced with random or directed libraries coding for 20 amino acid peptides.*

Subsequent machine learning on resultant positive clones revealed several features of interest, both predicted (selection for negatively charged amino acids) and not so very obvious very strong selection for tryptophan in the sequences of the artificial activation domains. Tests with directed library revealed a novel DW tandem repeat as providing maximum activity. To the best of my knowledge, such a sequence has never been demonstrated to regulate transcription with such high activity.

Overall, the study is well performed and the manuscript is well written. The screening method itself is not particularly novel in comparison to many other screening strategies. It appears to me, however, that the combination of the screen with data analysis could provide more insight into binding of IDPs to other molecules than more conventional analyses.

Response: We thank the reviewer for his/her enthusiasm on our work, and the clarity with which he/she has summarised our work. We decided to adapt a previously established screen to demonstrate the feasibility of the approach. We agree with the reviewer that there are interesting ways to design the screen and systems that one could investigate. We look forward to applying our method to other problems and helping the community to apply this framework to their own scientific problems that are amenable to this approach.

Comment 1.2: *I return again to the DW repeats. These are interesting for two reasons: First, while in vitro binding screens such as phage display have been shown to exhibit unusual amino acid biases, notably for abundances of tryptophan, I am not aware that in vivo screens show the same bias. This issue is discussed and the structural and thermodynamic consequences of multiple Trp tandem repeats were discussed and beautifully illustrated in the strange case of the "Trp Zipper", an extraordinarily stable beta hairpin discovered in a phage display screen (Cochran, A, et al. PNAS, 2001). Another unrelated structure that would be worth discussing is the co-crystal structure of the messenger RNA 5' cap-binding protein (eIF4E) bound to 7-methyl-GDP (Marcotrigiano, et al. Cell, 1997). Here the 7-methyl-GDP is nestled between two tryptophan rings forming an interesting and subtle interaction that might reflect something about the way that the author's weird TD repeat interacts.*

Response: The referee raises very interesting points and we agree that *in vivo* assays are unlikely to display amino acid bias. Unlike in phage display, negative selection is incorporated within IDR-Screen because the selection for function happens within the cell where a given sequence can encounter other proteins that are expressed. Moreover, we are selecting for transcriptional activity instead of strong binders to components of the transcriptional machinery. Thus, the *in vivo* screening step will likely select against sticky sequences. For these reasons, it appears less likely that IDR-Screen will exhibit unusual systematic/experimental amino acid biases. Nevertheless, functional sequences might still be systematically biased. We think as we perform different assays using this technology in the future, the existence of any systematic biases, if any, might become more obvious.

We thank the reviewer for pointing us to these two references that are very relevant to further our understanding of how the DW dipeptide repeats might behave and that they might be exploited in other systems. We have discussed and referenced the two papers in the manuscript to draw the parallel to other systems (Page 4; line 172).

Comment 1.3: *I'd suggest major revision to the figures, notably figure 1, 3a, b, d and 4a, b. These are not of the quality, elegance and clarity that I have come to expect from this group and make difficult to follow, what should be straightforward ideas. Figures 3b, d and 4b are the worst and need to be substantially simplified and clarified to provide clear information that a general reader can appreciate.*

Response: We appreciate the feedback on the general quality of the figures. Specifically, the schematics of the paper in Figure 1, Figure 3a and Figure 4a should have been of higher quality. We also agree that the feature importance tables in Figure 3b, Figure 3d and Figure 4b were overly complicated with too much details, making the major conclusions less clear to the readers.

We have split the figures, made them bigger and have also redesigned the schematics of Figure 1, old Figure 3a and old Figure 4a to make them of higher quality and clearer. Furthermore, we have simplified the feature importance description in **Figure 3**, **Figure 4B** and **Figure 5**, as well as provided clear descriptions of the features.

We hope that the revised manuscript and the figures meet the expectation of this referee for publication in MSB.

Reviewer #2:

Comment 2.1: *This manuscript describes a high-throughput method for finding what properties make a sequence activate transcription of a reporter gene when fused to a DNA-binding domain binding to the promoter of the reporter gene. Only a good handful of so-called activation domains are known from low-throughput experiments. This work represents so far the first high-throughput, unbiased screen of activatory peptides.*

Although a relatively low number of only 760 activating sequences were found in the screen, the authors were able to demonstrate in a cross-validation test that a machine learning classifier trained on a part of the activating and non-activating sequences can distinguish new activating and non-activating sequences (at 1:1 ratio) with a precision much higher than random expectation. The manuscript therefore makes a unique and valuable contribution to the very challenging topic of transcriptional activation.

Response: We would like to thank this referee for his/her enthusiasm on our work. We are also grateful for the extremely thoughtful, thorough and constructive suggestions to improve our work. Although the number of functional sequences of 739 (revised) is not very high, we believe it is astonishing that such a large number of completely random peptides can decide between life and death of the organism.

Comment 2.2: *According to the description in the Methods section, it seems that sequencing error rates are very high, because an error rate of 25% in the barcode are still accepted: 1 out of 4 nucleotides can be misread. That is an extremely high error rate and it would be important to comment on the causes and to discuss more the possible consequences for interpretation of the data.*

Response: We apologise for not being clear in our methods and for the potential misunderstanding. We wish to clarify that the sequencing error rates are not 25%. The error rates of sequencing in our experiments are very low, with more than 94% of sequencing cycles being over a score of Q30. Most importantly, to ensure high quality, we removed reads with *expected error rates* above 1 (i.e. more than 1 position based on the Q score of the sequence), further reducing the risk of having mistakes in the barcode assignments.

Response Figure 1: Representative Quality Score distribution of Sequencing Runs (screenshot from the MiSeq output). More than 94% of the sequencing cycles have a quality above Q30, which, together with the minimal Levenshtein distance (see next page) between barcodes of 4, reduces the concern that barcodes could have been wrongly assigned (please see below).

Furthermore, in order to accurately distinguish between replicates of samples (three replicates, barcoded), we have designed barcodes of length 8 and with minimum Levenshtein distances of 4 (i.e. a distance that consists in the number of substitutions, insertions and deletions required to change one nucleic acid sequence to another). This means that there would at least need to be 4 sequencing mistakes in order for any pair of the 3 barcodes that we used to be confused with each other. With this very stringent design in mind we have accepted error rates of up to 25% (only in the barcode region), which, for a barcode length of 8, translates to a maximum of 2 mistakes ($0.25 * 8 = 2$). Thus, the sequence reads obtained should be accurately grouped by their barcodes (specifically: AGGCAGAA, GGA CTCT and TAGGCATG). We have clarified this in detail in the Materials and Methods section (Page 10; Line 423).

Comment 2.3: *The authors describe that they cluster the sequences into clusters of similar sequences with a minimum sequence similarity threshold of 90%. It is not clear a priori why this should be necessary, as one would expect that in a random library sequences with different barcodes would be unique, and the chance of two sequences of length 15 residues to be the same by chance is on the order of $20^{(-15)} = 0$. The authors need to discuss the possible origin of these similar sequences.*

Response: The reviewer is right to point out that theoretically it is not necessary to select any threshold other than 100% identity to cluster reads into the distinct nucleotide sequences (centroids) present in the experiment. However practically, we picked this threshold as a way to balance the trade-off between over- and underestimating the number of “distinct” nucleotide sequences detected in the experiment.

To explain this further, let’s consider that the number of reads from the surviving sequences (i.e. they are selected for in the screen) across samples. Among these sequences, it is likely that a certain number of reads which pass our quality control filter may still contain errors. If we picked a

threshold of 100% identity, each of the distinct reads (some with errors) would be assigned its own centroid, whereas in reality some of the “unique sequences” that pass our filter might be a result of a same sequence with different sequencing errors, giving rise to the distinct reads (especially if their read counts are low).

This type of over-estimation of number of unique sequences would be especially detrimental to our downstream analyses, as it would artificially increase the number of activating sequences detected, whereas in reality they would have just been artefacts from the sequencing. On the other side of the trade-off, picking a loose threshold could lead to a different problem: that of under-estimation where truly different nucleotide sequences (with some similarity) would falsely be aggregated into a single centroid sequence.

However, as the reviewer also points out, since we are working with a random library, it is very unlikely for two independent nucleotide sequences to be similar to 90%. Therefore, our choice of a threshold of 90% will not be a problem.

Response Figure 2: The different unique sequences are shown in the sequence space (grey region) as dots. The size of the dots indicates the number of reads supporting the sequence. Points in the close proximity of sequences (circled region) with high read abundance (pink and purple) are grouped together into clusters if they are within a circle of radius of 90% sequence identity (indicated by grey arrows). The main sequence of the cluster is referred to as centroid sequences. This approach allows associating sequences with a few errors manifested in mismatches, insertions or deletions into clusters representing their “true” sequence. Given the vast sequence space in a random 60mer DNA library, it is very unlikely that two biological sequences from the experiment would be clustered together in this way (inter-cluster distances).

That is why we chose a more tolerant threshold for aggregation as (i) we do not run the risk of aggregating truly different sequences (preventing under-estimation of the number of sequences), whilst (ii) we make sure that sequences with errors are aggregated into the same centroids from which they originate (preventing over-estimating the number of sequences). In this way, we discovered 739 (centroid) sequences in the random library.

During the revision, we have ensured to use a clearer set of terminologies to describe these concepts and terms throughout the Materials and Methods section (Page 9; Line 413). Furthermore, we have created a schematic that helps to clarify these concepts, visualising the distinction between centroid sequences and the number of reads that support these sequences as a box in **Appendix Figure S2**.

Comment 2.4: Also, when the authors write that "The resulting mapping table for both the random and the designed libraries was aggregated to a count table where every sequence has its count recorded for each replicate", do they mean that every **centroid** has its counts recorded, which are the aggregate of all members of its cluster of similar sequences? Does it mean that 760 activating **centroid** sequences were discovered, or does the number 760 include the very similar sequences?

In the sentence referred to by the reviewer, we have used the wrong term “aggregate”, which has made this portion of description of the data processing unclear. The centroid sequences are not aggregated in any way. At this stage of the pipeline, the pre-processed sequencing reads (filtered and trimmed sequences which were used to build the clusters), are mapped back to the different centroid sequences. The number of reads that were uniquely mapped back to the centroid sequences in this way was recorded. This data was then used to estimate the growth rates (next section). The different centroid sequences are by definition different to each other and do not share a high degree of sequence similarity (less than 90%). In the later stages of the pipeline, where the centroid clusters are assigned to be functional or non-functional based on their growth scores, we do indeed have 739 very distinct functional sequences that do not share a high degree of sequence identity (see also response to **Comment 2.5**). This has now been clarified in the Materials and Methods section of the manuscript (Page 9; Line 413) as well as in **Appendix Figure S2**.

Comment 2.5: *The authors need to make sure that in their cross-validation benchmark on the machine learning predictors, none of the training sequences has a similarity to any of the test sequences higher than expected by chance. Otherwise, the prediction performance could be highly overestimated, as test sequences for which similar sequences have been trained on will be predicted much more accurately than the realistic cases in which no similar training sequence has been observed.*

The authors mention that many of the sequences they obtain are very similar to each other, hence the clustering. But how many are similar but have sequence identities below 80% or 90%? If pairs with sequence identities higher than what can be expected by chance (10-20%) are split between training and test set, the estimated performances of the predictors will be overestimated. A simple criterion could be to limit the sequence identity in a local alignment of the coding sequences to 30%. Here, sequence identity is defined as the fraction of identical aligned residue pairs to the length of the shorter sequence.

Response: We thank the referee for raising this point. Related to Comment 2.3 above and our calculations below, the peptide sequences from the random library are unlikely to be affected by the concern raised because their amino acid sequences are very different to each other (next paragraph).

To assess for similarity between peptide sequences, we did not use sequence identity as this assumes that the peptide sequences can be aligned. Whereas in our case, the peptide sequences are so different to each other that it is not possible to align them to a satisfactory degree. Instead we use the Levenshtein distance between sequences (also referred to in response to **Comment 2.2**). The Levenshtein distance in the amino acid sequence space between two sequences is the number of substitutions, insertions or deletions of amino acids required to transform one sequence into the other. We have calculated the pair-wise Levenshtein distances between all the functional peptide sequences of the random library and the wild-type TAD sequences in the design library. For each sequence, we have then selected the minimum distance to all other sequences, i.e. the distance between the pairs that are the most similar, and normalized by the sequence length.

The concern raised by the reviewer applies to a much lesser extent to the random library where the median of the minimum pair-wise normalised Levenshtein distance across sequences is 0.65 (**Response Figure 3**, Left). In other words, on average, 65% of the amino acids of the sequences need to be changed for two sequences to be identical.

Response Figure 3: The distribution of minimum pair-wise Levenshtein distances between functional sequences in random and design library.

For the design library, it is true that the sequences in the dataset by design are very similar as they are variants of a wild type TAD sequence. The wild-type sequence together with all its variants constitutes a TAD set (13 TAD sets in our study). It should be noted that the wild-type sequences themselves are significantly different (median of the minimum Levenshtein distance (normalized by length) among the sequence pairs is 0.65; see **Response Figure 3**, right). However, if we implement the constraint suggested by the referee, it would remove all point mutations and just retain one sequence per TAD set. Such a dataset would be too small for training and subsequent testing. Therefore, we did not originally incorporate this sampling strategy that controls for sequence similarity within TAD sets.

To address the reviewer's concern, we have now come up with a slightly revised strategy for the analysis of sequences in the design library. We have now applied *group* k -fold cross-validation ($k=5$) rather than k -fold cross-validation. A group is a TAD set, which is defined as a wild-type TAD sequence and all their variants. Therefore, for each data split, four-fifth of the 13 TAD sets (~10 TAD sets) were assigned to the training set and one-fifth (~3 TAD sets) were assigned as the validation set (this procedure was carried out 10 times). This ensured that variants within a TAD set (which are highly similar) were not split between the training and the validation sets. The results from this calculation are now presented in the manuscript.

We thank this reviewer again for raising this point. We feel that addressing this point has increased the robustness of the method and the models that we present in the paper. This has resulted in updating **Figures 4B** and **5** as well as **Appendix Figures S12** and **S13** and **Tables EV7** and **EV8**. This is also discussed in Materials and Methods (Page 12; Line 563) and **Appendix Figure S3**.

Comment 2.6: *There is a contradiction on page 18 in the online methods: "The dataset was split into 75% for training and the remaining 25% for testing..." This implies that the authors used 4-fold cross-validation, but in the next sentence they write "The entire machine learning process was performed with repeated cross-validation ($k = 5$ folds and 10 repeats) on the training set." The entire procedure of cross-validation and sampling is not clear, as the authors do not seem to have used the standard cross-validating procedure. For example what does "10 repeats" refer to?*

Response: We apologise for being ambiguous on what we refer to when using these terminologies. We have followed the definition as in *The Elements of Statistical Learning* (ISBN 978-0-387-84858-7) on page 222: "The training set is used to fit the models; the validation set is used to estimate prediction error for model selection; the test set is used for assessment of the generalization

error of the final chosen model. Ideally, the test set should be kept in a “vault,” and be brought out only at the end of the data analysis.” According to that definition, we first split the original data for the random library into a training set (75% of the sequences) and testing set (25% of the sequences). For the sequences in the training set, we have performed k-fold cross-validation ($k = 5$) and repeated that process a total of 10 times for training and validation of our models. We then took the top models from that procedure and evaluated performance on the testing set, which we kept separate during training (i.e. locked away in a data vault; 25% of the sequences from the original data as mentioned above). This allowed us to evaluate how the final models perform when totally new and unseen data is given.

Because of the new sampling strategy for the analysis of the designed sequences (please see response to comment 2.5), we now do not split the original data into an initial training and testing set due to the smaller number of TAD sets. Instead, we performed a *group* k-fold cross validation for training and validating the models (where a group is defined as a TAD set). The details of the sampling strategies applied in the revised manuscript for the analysis of the random, design and combined libraries are presented in **Appendix Figure S3** and are described in the Methods section.

We have clarified the definitions in the Materials and Methods section (Page 12; Line 543). We now clearly define what we refer to when using training, validation and testing sets in our machine learning approaches, and have made sure to be consistent when using these definitions across the manuscript. We have also improved **Appendix Figure S3** to make these distinctions clearer.

Comment 2.7: *Instead of demonstrating the usefulness of the learned features of activating domains using the sequence WDWDWDWD..., it would be more convincing to show the activation potential for a less obvious sequence where the authors make use of the specific minipatterns they have learned. The prediction of WDWDWDWD... could have been made even before because it is known from earlier work using site-directed mutagenesis (<http://www.pnas.org/content/111/34/E3506.full.pdf?with-ds=yes>) that mutations to W generally increase activation potential and that negative charges are important to prevent aggregation.*

Response: We agree with the reviewer that a clear link between W and activity could have been established based on the important work of Warfield *et al.* that we have also cited. However just repeating Ws or Ds did not function as TADs and we found it surprising and exciting that increasing the number of combined Ws and Ds would work comparably to VP16, one of the most active sequences. We fully agree that using the model as a generative tool to design new sequences and comprehensively testing their performance is a wonderful direction. We hope to undertake in the near future as a separate study.

Comment 2.8: *The description on top of page 17 seems to mean that the training ****and**** test sequences were subsampled to get a ratio of 1:1 between activatory and non-activatory sequences. However, the Supplementary Figures 6 and 9 show a dotted line with an "imbalance ratio" of 0.0116. This imbalance ration should be 0.5, shouldn't it? On the other hand, if test sequences were ***not*** subsampled, I would expect an imbalance ratio of $760 / 67000 = 0.113$ and not 0.116.*

Response: We apologise for not being clear. We aimed to maintain a similar ratio of imbalance between the functional:non-functional sequences (complete dataset: $739/63385=0.0117$) in the training (75%) and the testing (25%) set for the random library. The actual value after sampling may deviate slightly from this ratio as observed for the training set (0.0115). While performing the k-fold cross validation, we subsampled the training set to the minority class to ensure that the model is not biased by the majority class. However, while evaluating the performance on the validation set or the testing set, the imbalance is maintained to prevent over-estimation of performance and reflect the real imbalance in the original data set. The reviewer pointed out correctly that the number of non-functional sequences in the main text is mentioned as 67,000, which was incorrect (this has been revised and corrected now).

We have now described at what stage the sub-sampling is performed in the Materials and Methods section (Page 12; Line 550) as well as highlight it in **Appendix Figure S3**. We also provide the accurate number of sequences used in the study throughout the manuscript in several instances (related to last comment below).

Comment 2.9: *Figures S6 and S9: The x axis labels "1 - sensitivity" should be "1 - specificity". The FPR needs to be defined explicitly in the online methods.*

Response: We now clearly defined how the false positive rate (FPR) in the machine learning stages is calculated in the Materials and Methods section (Page 14; Line 665). We apologise for this error and have them errors in the x-axis labels of **Appendix Figure S6, S8, S12, S13** to "1-specificity".

Comment 2.10: In the main text (Page 3) the authors write "...we obtained robust measurements for 67,000 variants." In the methods, the authors write "Given the imbalance in our dataset (760 functional: 65,517 non-functional...)". This adds up to 66270, not 67000. If the authors want to round, it should be 66000.

Response: We apologize for making this mistake. We have now made sure to give the correct number for the random library analysis: a total of 64,124 sequences used in the study (63,385 + 739; non-functional and functional sequences, respectively). The reason for revision of the numbers is because do not include the stop codon sequences from the dataset (i.e. sequences with a stop codon as its first trinucleotide). This change is reflected at several instances throughout the manuscript.

In summary, addressing the constructive comments of this referee has greatly strengthened the manuscript. We hope the referee finds the revised manuscript suitable for publication in MSB.

Reviewer #3:

Comment 3.1: *Ravarani et al. present an integrated experimental-computational analysis of intrinsically disordered regions, in particular, the transactivation domain of HSF1. They start with an elegant screening system that inserts a random DNA sequence instead of the wt TAD of HSF1; they then apply heat shock, which leads to death in absence of a functional TAD. They thus discover a few hundred functional transactivation domains and develop a machine learning algorithm to discriminate functional from non-functional sequences. They supplement this by screening TAD of other transcription factors, including mutants thereof and obtain a somewhat better predictor.*

Response: We thank the reviewer for his/her enthusiasm of our work and for the constructive comments on our work. Addressing the issues raised has improved the quality of our work and manuscript. We hope that the revised manuscript meets the expectation of this referee for publication in MSB.

Comment 3.2: *This work is the first attempt to examine functionality of non-motif disordered regions (as far as I am aware) using a large-scale approach and should thus be of interest to a large audience. Their main result (that a large fraction of sequences can work as TADs) is interesting and novel. However, I personally think that the manuscript focusses a bit much on the machine learning aspect, which, while performed and discussed extensively, I find less insightful and prediction performance remains relatively low.*

Response: We appreciate this point of the referee. We agree that the field of transcription initiation is of fundamental importance and that this has guided us to choose it as our system of study. Firstly, we hope that the proposed framework will not be limited to studying transactivation domains, and that it can be applied to a range of systems where IDRs can be screened for a defined function. The generality and the emphasis on the machine learning part of the paper were aimed at making this point.

Secondly, while it seems that the predictive power of the models that were developed are modest (for which we give additional reasons below), it allows one to compare, extract and interpret which of the defined and definable features are important for TAD functionality in a common, comparative framework. For these reasons, we respectfully feel that it is worth discussing the machine learning part to the extent that we did in our manuscript.

Comment 3.3: *In the initial screen, the authors massively under sample sequence space (measuring about 10^5 out of 20^{20}). I'm hence very surprised that they obtain any functional sequences at all, let alone that many (~1% of the sequences screened). That alone is, I think, a very interesting result*

- the authors do not seem to comment on it further. It does go to show just how "fuzzy" and non-specific the TAD interactions really are, and I think it deserves further discussion and examination. To illustrate how much 1% really is, compare it to e.g. results from phage display for peptide binding motifs. Out of large randomized libraries (around 10^{10}), one usually obtains a handful of binders (so far less than even 0.01%). Surely, even all sequences containing the mini-motif (DE in proximity to FWY) would not make up 1% of all sequences?

Response: We were also surprised to note that 1% of the sequences were functional and agree that is a very large number, considering that a short and completely random sequence can decide between life and death of the organism hosting it. From the analysis of the structures of TADs in complex with co-factors (**Appendix Figure S9**), it is clear that different sequences adopt similar binding mode when interacting with the various co-factors. This suggests that a large number of sequences can and are compatible with co-factor interaction. Furthermore, many sequences can fulfil the role of a TAD by interacting with one of many components of the transcriptional machinery and hence a larger fraction of the sequences may survive the screen.

From a technical point, unlike phage display, in the case of IDR-Screen, the random peptide is not selected for binding to a specific protein of the transcriptional machinery but to any of the over 100 proteins that are involved in transcriptional initiation (selection is for function, and not explicitly for binding). Therefore, we might pick up more sequences that are functional through IDR-Screen compared to phage display for identifying binders to a specific protein of interest. We thank the referee for encouraging us to discuss this point, which we do now in the discussion of the revised manuscript (Page 7; Line 313).

Comment 3.4: *Going back to the first point (non-specificity of TADs), I found it a bit surprising to see so much "red" in Figure S8, even in the alanine scanning. So many single point mutations leading to non-functional TADs would imply more specificity, which is surprising. From eye-balling Fig S8, it seems that most TADs fall into two camps: highly sensitive (e.g., AH or Gln3), tolerating only very few mutations, or are highly permissive (vP16 or Oaf1). From the first point, I would've expected all TADs to be of the second camp (remember, 1% of random sequences are functional, even single mutants from a known sequence are a tiny slice of sequence space). This certainly warrants further analysis. Are there technical/experimental reasons for this difference between different TADs?*

Response: We thank the referee for this comment. Motivated by this suggestion, we have calculated the tolerance score for every TAD in our design library. Tolerance is defined as the number of variants that confer survival over all variants tested for that TAD. We then analysed the distribution of the tolerance scores of the different TAD sequences. Intriguingly, we find a unimodal distribution where a majority of the TAD sequences have intermediate tolerance scores. As the reviewer points out, VP16 and Oaf1 are at the most tolerant end of the spectrum whereas Gln3 is in the least tolerant end of the spectrum. This suggests that most TADs are tolerant to a certain extent and this is likely to be determined by the nature of the substitution as summarised in an updated version of **Figure 4A**. We are not aware of specific technical or experimental reasons for this difference between the TADs.

We now present an analysis of mutational tolerance for the different sequences and present the results as **Appendix Figure S11** and discuss it in the main text (Page 6; Line 247).

Response Figure 4: Histogram of the tolerance for scanning (point mutation) variants. The tolerance values were grouped into 6 bins and each group is coloured according to its tolerance (gradient between green, most tolerant and red, least tolerant). The name of the sequence set as well as the WT sequence are provided.

Comment 3.5: The raw sequencing data does not seem to be given, only the author's assignment of "functional" and "non-functional" even in the supplement, the authors should provide the raw data, or at least their growth estimate. Such experiments tend to be noisy, so many sequences are likely not real. Did the authors not try to clone a few identified sequences and measure growth under heat shock? (except for the handful in Fig 4c).

Response: We thank the referee for raising this point. We now provide all the growth estimates in supplementary tables. The reviewer is right to point out that these experiments tend to be noisy. Apart from characterization by the high through-put selection and sequencing experiment, we did initially pick a few single colonies, performed the dilution experiment (below) and Sanger sequencing to confirm the behaviour of some of the sequences, which were not presented in our original manuscript. This is now discussed (Page 3; Line 137) and presented as **Appendix Figure S4** in the revised manuscript. Furthermore, we now provide the estimated growth rates in **Table EV1** and **Table EV5**.

Figure 5: Spot dilution assay for sequences identified during the screen of the random library both at permissible temperature for Δ HSF1 strains to grow (30°C) and at non-permissible heat shock

temperatures of 37°C. VP16 and Δ TAD-HSF1 strains are given as reference points. Growth estimates (in a.u.) and classification are shown on the right.

Comment 3.6: *In the same vein, I see that a number (25 or ~3.5%) of sequences are a single amino acid, with another 100 or so (>10%) being 4 amino acids or less. These are surely false positives? This would translate to a FP rate far higher than the authors suggest.*

Response: In the earlier version of the manuscript, the FPR metric was defined using growth rate estimates based on negative sequences that we knew will not confer TAD functionality from prior knowledge (e.g. those containing a stop codon as the first trinucleotide). We felt that this was an objective criterion. However, we do agree that this metric does not reflect the experimental FPR (i.e. sequences that were wrongly detected as functional during the selection experiment). For instance, as the reviewer points out, sequences with less than 4 amino acids are less likely to be functional and are unlikely to be true positives. By considering all sequences with 4 or fewer amino acids as non-functional we obtain the estimated false positives as 13.5% (100 sequences with 4 or fewer amino acids / 739 sequences). We now discuss this point in the Materials and Methods (Page 11; Line 488) to ensure that the reader is aware of these considerations.

Comment 3.7: *I apologize for making a relatively straightforward technical point, but it appears to me that the authors do always evaluate their ML frameworks by cross-validation (hence the performance of the ones trained on "designed sequences" performs better than both the ones trained on the random-screened sequences and the combined one). To obtain a better and more comparable evaluation it would make sense to instead use a separate validation set (not used for training) to ascertain performance. Would also help to see how much the combined ML actually performs better than the one trained on the results from the screen on random sequences.*

Response: We apologize for the ambiguity and for the misunderstanding. In the previous version, we indeed performed the calculations according to what the referee mentions. For all models that were trained, we did have an initial training (75%) vs. testing (25%) set split of the data where the testing set was never used in training the model. Then according to the k-fold cross-validation, the training set was further split into a training set and validation set, with which the models were trained and evaluated. The final model was then evaluated with the initial testing set which was never used for training. This is what we reported in the paper.

We did the same for the *random* library in the revised manuscript. On the other hand, to minimize sequence similarity between the training and validation sets in the *design* library, we have now come up with a slightly revised strategy. We have applied *group* k-fold cross-validation (k=5) rather than k-fold cross validation. Due to the large number of sequence sets that this approach requires, we have just used cross-validation for the design library analysis without the additional evaluation of a test set due to limited size of the dataset upon splitting. For the combined library, we have implemented the reviewers suggestion and evaluated the models trained with the combined library on the testing set of the random library. We note that the performance remains comparable to the models trained purely based on the random library.

We have now used consistent terminology throughout the manuscript and clarified this in the Materials and Methods section of the manuscript as well as in the **Appendix Figure S3**. The results from this calculation are now presented in the manuscript in updated **Figures 4B** and **5** as well as **Appendix Figures S12** and **S13** and **Tables EV7** and **EV8**. This is also discussed in Materials and Methods (Page 12; Line 550) and **Appendix Figure S3**.

Comment 3.8: *This may be somewhat a question of personal preference, but I would think a more descriptive title is better suited (as the authors do not actually test "functional disordered protein regions" in general, but rather focus on the rather specialized case of transcription factor transactivation domains).*

Response: Thank you for this suggestion, we have changed the title to the following "High-throughput discovery of functional disordered protein regions: investigation of transactivation domains". We hope this new title reflects the generality of the approach and the specific system (transactivation domains) that it was applied to.

Comment 3.9: *In Fig. 4c) I do not see cells with WD10 sequences survive "much better" than wt HSF... A little bit better, if that. Some better or clearer data (longer growth times, etc.) would be needed to make that claim.*

Response: We agree that the comparatively increased growth as inferred from the dilution assay is modest. We rephrase our statement to indicate that the that WDx10 sequence performs comparably to the wild-type sequence and added the results to **Figure 6**.

Comment 3.10: *It might be more instructive (and impressive) to try a number of other TAD sequences designed with the discovered features (rather than just all W, all D and WD10).*

Response: We fully agree that using the model as a generative tool to design new sequences and comprehensively testing their performance is a wonderful direction. We hope to undertake in the near future as a separate study.

2nd Editorial Decision

3 April 2018

Thank you for sending us your revised manuscript. We have now heard back from the referee who was asked to evaluate your study. As you will see below, reviewer #2 is satisfied with the modifications made and thinks that the study is now suitable for publication.

Before we formally accept the manuscript, we would like to ask you to address some remaining editorial issues listed below.

REVIEWER REPORT

Reviewer #2:

The authors have responded very constructively to the reviewer comments and have substantially improve their manuscript. All major issues have been addressed in my view.

Corresponding Author Name: Charles Ravarani, Alexander Erkin and M. Madan Babu

Manuscript Number: MSB-18-8190